# Metabolic rewiring of macrophages by epidermal-derived lactate promotes sterile inflammation in the murine skin

Uttkarsh Ayyangar [1,2]✉, Aneesh Karkhanis [3], Heather Tay[3], Aliya Farissa Binte Afandi[3], Oindrila Bhattacharjee[1], Lalitha KS[4], Sze Han Lee [3], James Chan[3,5] & Srikala Raghavan [1,3]✉

## Abstract

Dysregulated macrophage responses and changes in tissue metabolism are hallmarks of chronic inflammation in the skin. However, the metabolic cues that direct and support macrophage functions in the skin are poorly understood. Here, we show that during sterile skin inflammation, the epidermis and macrophages uniquely depend on glycolysis and the TCA cycle, respectively. This compartmentalisation is initiated by ROS-induced HIF-1α stabilization leading to enhanced glycolysis in the epidermis. The end-product of glycolysis, lactate, is then exported by epithelial cells and utilized by the dermal macrophages to induce their M2-like fates through NF-κB pathway activation. In addition, we show that psoriatic skin disorder is also driven by such lactate metabolite-mediated crosstalk between the epidermis and macrophages. Notably, small-molecule inhibitors of lactate transport in this setting attenuate sterile inflammation and psoriasis disease burden, and suppress M2-like fate acquisition in dermal macrophages. Our study identifies an essential role for the metabolite lactate in regulating macrophage responses to inflammation, which may be effectively targeted to treat inflammatory skin disorders such as psoriasis.

**Keywords** Sterile Inflammation; Epithelial-immune Crosstalk; Lactate Metabolism; Metabolic Compartmentation; Psoriasis
**Subject Categories** Immunology; Metabolism; Skin

## Introduction

Inflammation is an orchestrated phenomenon associated with increased immune cell burden, enhanced synthesis of cytokines and chemokines, and metabolic reprogramming of the tissue. Interestingly, growing evidence suggests that inflammatory diseases are associated with an increase in glucose uptake and augmentation of glycolysis by disease-associated cell types (Soto-Heredero et al, 2020). The increase in glycolysis not only meets the rapid bioenergetic demands of the disease-associated cell types but also aids in generating biosynthetic precursors for synthesizing effector molecules (O'Neill et al, 2016). This phenomenon has been observed in inflammatory disease conditions like cancer and rheumatoid arthritis (Chang and Wei, 2011; Liberti and Locasale, 2016). Conceivably, increased glucose uptake and glycolysis in one compartment of the organ during inflammation must lead to alteration in the metabolic milieu for other resident cell types. These alterations can have directive effects on the functional state of other cell types in the tissue, especially immune cells that are newly recruited into the tissue during inflammation. As metabolites have been shown to function as signaling molecules, changes in their relative levels in the tissues can direct the functional state of immune cells (Chouchani, 2022; Figlia et al, 2020; Baltazar et al, 2020).

We and others have shown that macrophages, an innate immune cell types, are the earliest responders to inflammatory cues in tissues (Kurbet et al, 2016; Bhattacharjee et al, 2021; Delavary et al, 2011). These macrophages exacerbate inflammation by acquiring an M2-like fate that cause exacerbated degradation the extra-cellular matrix by increased MMP synthesis. Interestingly, several recent studies have revealed that macrophages exhibit a high degree of metabolic plasticity to fulfill effector functions (Russell et al, 2019; Saha et al, 2017; Van den Bossche et al, 2017). In vitro, macrophages can acquire distinct pro-inflammatory M1 and pro-remodeling M2 fates. These M1 and M2 states of the macrophages are associated with unique metabolic programs where M1 macrophages preferentially depend on glycolysis while the M2 macrophages depend on tricarboxylic acid cycle (TCA) and oxidative phosphorylation (OXPHOS) (Russell et al, 2019; Saha et al, 2017; Van den Bossche et al, 2017). However, how the complex metabolic milieu of inflamed tissues directs macrophage effector function in vivo is not well understood (Wculek et al, 2022).

Interestingly, recent reports suggest that in inflammatory skin disease conditions such as psoriasis and atopic dermatitis, are also associated with increased glucose uptake in the epidermis (Cibrian et al, 2020; Zhang et al, 2018; Choi et al, 2020). Notably, inhibition

[1]Centre for Inflammation and Tissue Homeostasis, Institute for Stem Cell Science and Regenerative Medicine, Bangalore, India. [2]School for Chemical and Biotechnology, Sastra University, Thanjavur, India. [3]A*Star Skin Research Labs, Agency for Science, Technology and Research, Singapore, Singapore. [4]Animal Care and Resource Centre (ACRC), National Centre for Biological Sciences (NCBS), Bangalore, India. [5]Singapore Institute of Food and Biotechnology Innovation, Agency for Science Technology and Research, Singapore, Singapore. ✉E-mail: uttkarsha@instem.res.in; Srikala_Raghavan@asrl.a-star.edu.sg

of glucose uptake by the epidermal compartment leads to a remarkable reduction in skin inflammation which is accompanied with reduction in macrophage burden (Zhang et al, 2018; Choi et al, 2020). However, how might the epidermal metabolic reprogramming shape the effector functions of dermal macrophages in inflammatory skin conditions is not well understood.

In this study, using a mice embryonic model of skin inflammation (epidermal integrin β1 conditional KO mice), we report that the epidermis and macrophages augment unique yet complimentary metabolic programs where the epidermis augments glucose uptake and glycolysis and the macrophages augment TCA cycle. This metabolic program is initiated by an early increase in reactive oxygen species (ROS) that aids in enhanced stabilization of glycolysis regulator, HIF1α (Hypoxia Inducible factor), in the epidermal compartment. Enhanced glycolysis in the epidermis leads to increased generation of glycolysis-end-product lactate that is subsequently exported and utilized by macrophages in the dermal compartment to augment a pro-remodeling fate that is characterized by increased MMP9 generation. Notably, inhibition of glycolysis and its regulators in epidermis, TCA cycle in macrophages, and lactate-mediated crosstalk between the two compartments led to a remarkable reduction in the pro-remodeling fate acquisition in macrophages and in turn, skin inflammation. Mechanistically, we show that lactate augments TCA cycle and MMP9 generation, in part, through NF-κB activation. Finally, we report a similar lactate-mediated crosstalk between epidermis and macrophages in imiquimod induced mouse model of psoriasis-like dermatitis. Remarkably, inhibition of this crosstalk led to a significant reduction in psoriasis burden, thereby indicating the translatability of our findings to disease states. Taken together, our results underpin lactate metabolism as an important principle that can be targeted to treat inflammatory skin disorders.

# Results

## Epidermal inflammation in embryonic skin is associated with increase in glucose uptake and glycolysis

As reported previously, loss of integrin beta1 (itgβ1) from the epidermal compartment in mouse embryonic skin (cKO), leads to an augmentation of an inflammatory response characterized by increased macrophage cell infiltration and pro-remodeling fate acquisition (Kurbet et al, 2016; Bhattacharjee et al, 2021). However, the key metabolic pathways augmented by the epidermal compartment upon loss of itgβ1 remains to be uncovered. To gain insights into the metabolic state of the epidermal compartment, we used a previously generated NGS data from the epidermis, macrophages, and fibroblasts of WT and itgβ1 cKO embryonic skin and identified differentially expressed metabolic pathways in cKO epidermis (Bhattacharjee et al, 2021). Pathway analysis and qPCR validation suggested an increase in the expression of genes associated with glucose uptake and glycolysis metabolism in the cKO epidermis (Fig. S1A–C). Immunofluorescence (IF) staining of the cKO skin with GLUT1 (Glucose transporter 1) and LDHa (Lactate dehydrogenase A) and western blot analysis of HK2 (Hexokinase 2) suggested a specific increase in the expression of glucose transporter and glycolytic enzymes in the cKO epidermis compared to WT (Fig. 1A–D). We next performed steady-state metabolomics

to quantify relative changes in the levels of glycolysis metabolites in the epidermal compartment of the cKO skin. Our results suggested a global increase in the levels of glycolysis intermediates and end product, lactate in cKO epidermis compared to WT (Fig. 1E,F). The KO of epidermal integrin β1, under the KRT14 promoter, is initiated around embryonic day E12.5/13.5. We, therefore, set out to examine when the glycolytic reprogramming is set up developmentally. Temporal analysis of the expression of epidermal GLUT1 expression in the cKO skin suggested a significant increase in epidermal GLUT1 expression only at embryonic day E18.5 (Fig. 1G).

Glycolytic end products pyruvate and lactate feed directly into the TCA cycle as part of the central carbon respiration chain. We next examined the expression of TCA cycle genes in the epidermal compartment from WT and cKO epidermis. Interestingly, NGS and qPCR analysis suggested a general reduction in all the TCA cycle enzyme transcripts in the cKO epidermis (Fig. S1D,E). IDH1 (Isocitrate dehydrogenase 1) and CS (Citrate Synthase) IF staining suggested a reduction in the expression of TCA cycle enzymes in the cKO epidermis (Fig. 1H–J). Furthermore, steady-state metabolomics confirmed a decrease in levels TCA cycle metabolites in E18.5 cKO epidermis compared to WT (Fig. 1K). Taken together, our results suggest that loss of epidermal integrin β1 leads to a "glycolytic switch" in the epidermal compartment that leads to enhanced lactate generation in the cKO the epidermal compartment (Fig. 1L).

## Enhanced epidermal HIF1α stabilization directs the glycolytic switch in the embryonic skin epidermis during inflammation

We next aimed to identify the molecular events that initiated the metabolic reprogramming in the cKO epidermal compartment. HIF1α regulates glycolysis under both hypoxic and normoxic conditions (McGettrick and O'Neill, 2020). Interestingly, the NGS analysis suggested enrichment in pathways associated with response to hypoxia (Fig. 2A). This was associated with the upregulation of genes transcriptionally regulated by HIF1α (Fig. S2A). Consistent with this, we observed an increase in the expression of HIF1α and its downstream targets such as—*Serpina1, Vegfa, Timp1, Fn1, Id2, Krt14, Ptgs2,* and *Igfbp3* in the cKO epidermis compared to the control using IF and qPCR quantification (Figs. 2B–D and S2B,C). Temporal HIF1α expression analysis using IF further suggested augmentation of HIF1α expression in the cKO skin as early as embryonic day E17.5 (Fig. 2C). These results suggest an increase in epidermal HIF1α expression in cKO skin that precedes epidermal "glycolytic switch" at E18.5.

We next asked if increased HIF1α expression is directly associated with glycolytic upregulation in the epidermal compartment. To address this, we treated pregnant dams harboring WT and cKO embryos with inhibitor of HIF1α dependent transcription, chetomin (3 mg/kg)(Kung et al, 2004) (see Methods section, Fig. S2D). The inhibition of HIF1α using chetomin lead to a significant decrease in the transcription of glycolytic enzymes, expression of HIF1α targets—COX2 and KRT14, and expression of GLUT1 and LDHa in the cKO epidermis compared to DMSO treated cKO controls (Fig. 2E–I). These results suggested that the "glycolytic switch" in the epidermal compartment is augmented due to enhanced epidermal HIF1α expression and stabilization (Fig. 2J).

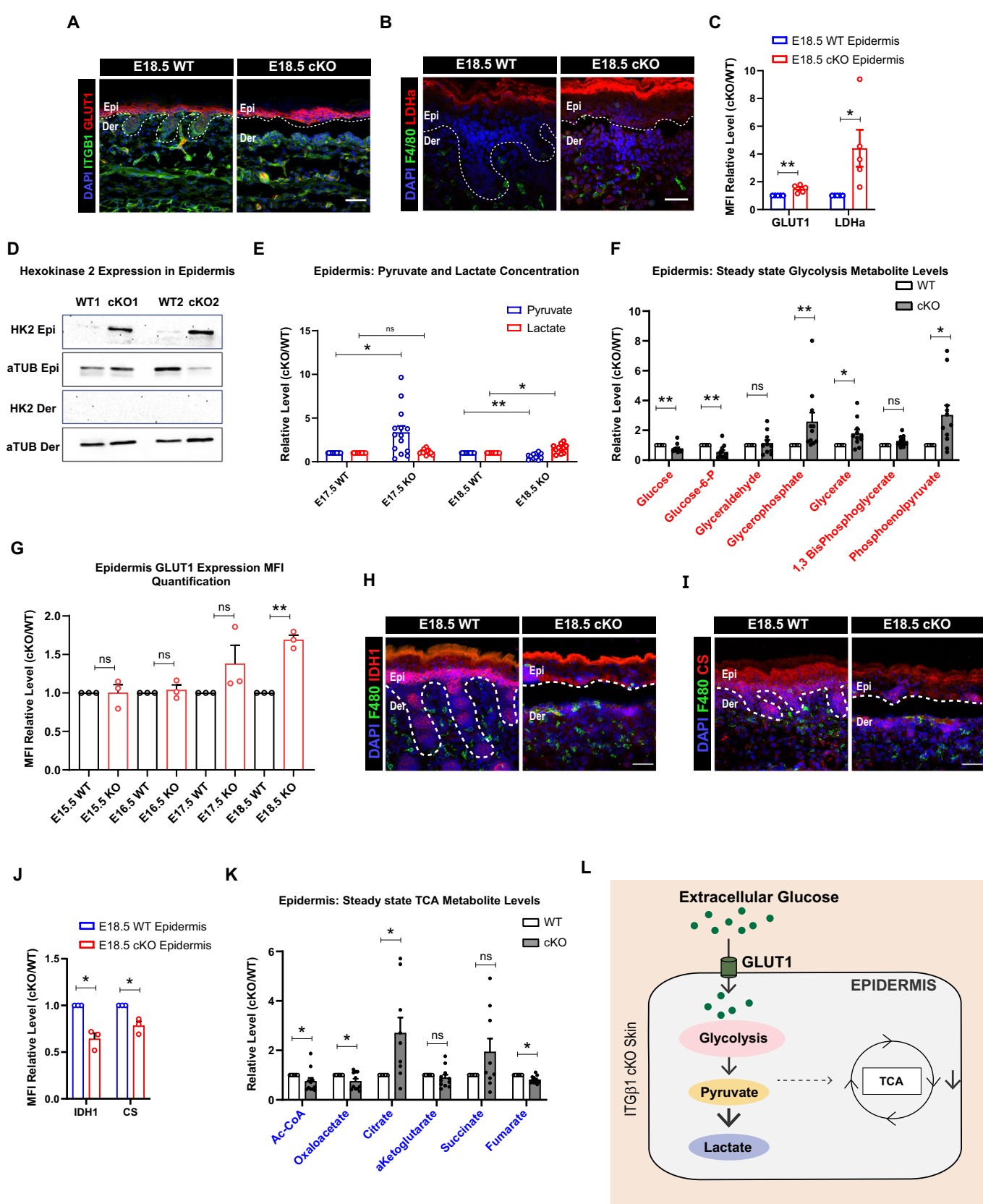

**Figure 1.  Increased glycolysis and decreased TCA in epidermal compartment in itgβ1 cKO skin.**

(A–C) Epidermal compartment of cKO skin showing increased expression of GLUT1 ($N = 5$) (A), LDHa ($N = 4$) (B) quantified in (C). (D) Western blot for Hexokinase 2 (hk2) in epidermal and dermal compartment of cKO skin (D). (E) Relative concentrations of pyruvate and lactate from E17.5 and E18.5 epidermis isolated from WT and cKO skin quantified using steady-state metabolomics (E) ($N = 5$). (F) Steady-state metabolomic quantifications for Glycolytic metabolites from epidermis of E18.5β1 cKO and WT skin (F) ($N = 5$). (G) MFI quantification of relative GLUT1 expression in cKO compared to WT epidermis from embryonic day E15.5, E16.5, E17.5, and E18.5 ($N = 3$). (H–J) Epidermal compartment of the skin showing reduced expression of IDH1 (H) ($N = 4$), CS (I) ($N = 3$) quantified in (J) cKO compared to WT. (K) Steady-state metabolomic quantifications of Relative levels of TCA cycle metabolites in epidermal compartment of cKO and WT skin (J) ($N = 5$). (L) Schematic showing glycolysis and TCA cycle changes as suggested by transcriptomic, proteomic, and metabolomic approaches in the cKO skin compared to WT. Data Information: All the microscope images (A, B, H, I) were quantified (C, G, J) in ImageJ software. The number of biological replicates ($N$) is given for each panel in the figure legend. Scale bars: 50 μm. Student's t-test was performed for statistical analysis. *$p \leq 0.05$, **$p \leq 0.01$, ns = not significant (student's t test). All graphs are presented as mean ± SEM. WT experimental values are normalized to 1 for relative quantification. Source data are available online for this figure.

## Enhanced ROS generation stabilizes HIF1α in the Itgβ1 cKO skin epidermis during inflammation

We next aimed to identify the mechanism of HIF1α stabilization in the cKO epidermis. Several studies have identified a role for ROS generated during trauma induced injury and tumors in stabilizing HIF1α response (Mittal et al, 2014; Dunnill et al, 2017). GSEA analysis of the cKO epidermis suggested enrichment in pathways associated with an active response to increased ROS (Fig. 3A). Temporal analysis of cKO and control skin using IF staining of 8-OHdG, (ROS indicator), suggested increase in ROS expression at embryonic day E16.5 cKO epidermis that quenches by E18.5 (Fig. 3B,C). Interestingly, we observed increased transcription of ROS source genes—*Alox5, Alox8, Duox1, Duoxa1, Mpo*, and *Xdh* even at embryonic day E18.5 (Fig. 3D). We, therefore, reasoned that the increased ROS burden in the cKO skin might be counter-balanced by increased expression of ROS scavengers in the cKO epidermis. Consistent with this, we observed increased transcription of ROS scavengers—*Gpx3, H6pd, Nqo1*, and *Prdx6* in E18.5 cKO epidermis (Fig. 3E). IF staining further suggested a global increase in the expression of global glutathione and catalase in the cKO skin compared to WT (Fig. 3F). These results suggest that loss of epidermis itgβ1 leads to an early increase in ROS levels that is subsequently quenched by increased levels of anti-oxidants in the skin.

We next aimed to identify the potential causes for the early increase in the ROS levels in the cKO skin. We previously showed that the loss of epidermal integrin β1 leads to the detachment of the epidermal compartment from the underlying basement membrane. Increase in ROS upon ECM detachment has been reported previously in metastasizing cancer cells (Mason et al, 2016; Davison et al, 2013; Schafer et al, 2009). We thus reasoned that ECM detachment increases mechanical stress in the epidermal compartment that, in turn, causes increased levels of reactive oxygen species. Notably, IF staining additionally suggested decreased expression of hemidesmosome integrin β4 at embryonic day E16.5, further exacerbating the mechanical stress in the cKO epidermis. We confirmed the increase in the epidermal mechanical stress at embryonic day E16.5 by observing increased expression of epidermal stress keratin, KRT6, and mechanical stress-induced ECM component, TNC (tenascin C) in cKO skin compared to WT (Fig. S3A). These results suggest that mechanical stress caused by loss of epidermal integrin β1 and aberrant expression of integrin β4 potentially leads to increased ROS generation in the cKO skin.

Since our temporal analysis suggested that ROS augmentation (at embryonic day E16.5) precedes HIF1α stabilization (at embryonic day E17.5), we reasoned that ROS might be the potential activator for HIF1α signaling in the cKO epidermis. To address this, we treated pregnant dams harboring WT and cKO embryos with the ROS scavenger, N-acetyl cysteine (NAC) (Zafarullah et al, 2003) (Fig. S2D). Interestingly, NAC treatment led to reduced expression of 8-OHdG and HIF1α in the cKO epidermal compartment compared to the PBS treated controls (Fig. 3G,I). IF staining further suggested a significant reduction in expression of HIF1α targets—KRT14, COX2, GLUT1, and LDHa in the cKO epidermis treated with NAC compared to the controls (Figs. 3H,J and S3B,C). Taken together, temporal analysis combined with pathway inhibition results suggest that an early ROS-HIF1α axis directs "glycolytic switch" in the cKO epidermis (Fig. 3K).

## Macrophages are potential sinks for lactate during sterile inflammation in embryonic skin

As described previously, enhanced glycolysis in the epidermal compartment led to lactate accumulation. We next aimed to understand the fate of epidermally generated lactate. Recent in vitro studies have suggested that the increase in lactate accumulation within the cells lead to increased membrane localization of lactate exporter MCT4 (monocarboxylic acid transporter 4) (Benjamin et al, 2018; Kirk et al, 2000; Miranda-Gonçalves et al, 2013). IF staining of the cKO epidermis at E17.5 and E18.5 showed an increase in the membrane expression of MCT4 transporters (Fig. 4A). To identify which population of epidermal cells, have membrane expression of MCT4, we performed MCT4 co-IF with basal keratinocyte marker KRT5. MCT4 co-staining with KRT5 suggested that basal cell population in the epidermis primarily express membrane MCT4 (Fig. S4A). These data, in conjunction with steady-state metabolomics (Fig. 1E), suggest that glycolysis end product lactate is generated in the epidermal compartment of cKO skin and subsequently exported potentially from the basal keratinocytes.

We next aimed to identify the role of epidermally exported lactate in cKO skin. We previously showed that the loss of epidermal integrin β1 leads to increased infiltration and activation of dermal macrophages that, in turn, generate matrix metalloproteinases that degrade the basement membrane (Kurbet et al, 2016; Bhattacharjee et al, 2021). In addition, previous studies have suggested a potential role for lactate in driving macrophage effector responses in inflammatory disease conditions (Yang et al, 2020; Li et al, 2024; Noe et al, 2021; Kes et al, 2020). We, therefore, reasoned that the macrophages may serve as potential 'sinks' and/or 'sensors'

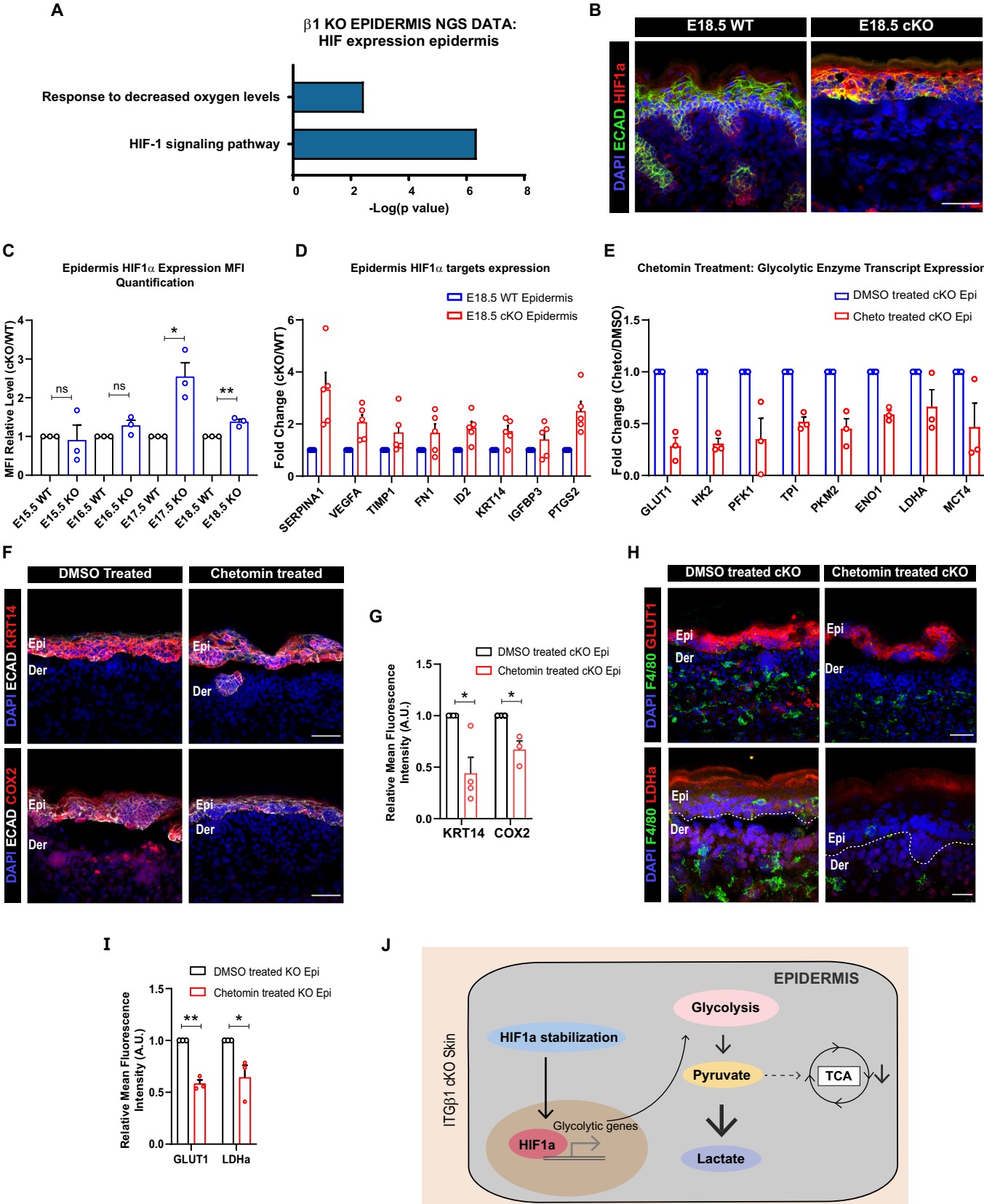

**Figure 2.   Increased HIF1α in epidermal compartment in itgβ1 cKO skin drives glycolytic switch.**

(A) NGS analysis of E18.5 cKO and WT epidermis showing enhanced clustering in HIF1 signaling pathway ($N = 2$). (B) Epidermal compartment of cKO skin showing increased expression of HIF1α in E18.5 cKO and control skin (B) ($N = 3$). (C) MFI quantification of relative HIF1α expression in cKO compared to WT epidermis from embryonic day E15.5, E16.5, E17.5, and E18.5 ($N = 3$). (D) qPCR showing increased transcription of HIF1α targets in the cKO epidermis compared to WT ($N = 5$). (E) qPCR showing reduced transcription of HIF1α targets in the cKO epidermis treated with HIF1α inhibitor, chetomin, compared to DMSO-treated controls ($N = 3$). (F, G) Epidermal compartment of the cKO skin showing reduced expression of HIF1α targets KRT14 ($N = 4$) and COX2 ($N = 3$), quantified in (G) in chetomin treatment compared to DMSO treated controls. (H, I) Epidermal compartment showing decreased expression of GLUT1 ($N = 3$) and LDHa ($N = 3$) quantified in (I) in chetomin, treated cKO skin compared to DMSO-treated cKO skin controls. (J) Schematic showing glycolysis regulation by HIF1α in cKO skin. Data Information: All the microscope images (B, F, H) were quantified (C, G, I) in ImageJ software. The number of biological replicates (N) for each panel is given in the figure legend. Scale bars: 50 µm. Student's t-test was performed for statistical analysis. *$p \le 0.05$, **$p \le 0.01$, ns = not significant (student's t test). All graphs are presented as mean ± SEM. WT experimental values are normalized to 1 for relative quantification. Source data are available online for this figure.

for epidermally-derived lactate during sterile inflammation in the cKO skin. NGS analysis of the macrophage compartment suggested an increase in the pathways associated with glucose deprivation (Fig. S4B). This was further associated with downregulation of several glucose transporters in the macrophages in the cKO skin compared to WT (Fig. S4C). This suggested a reduction in the glucose-dependent metabolic program in the macrophages in the cKO skin. Surprisingly, qPCR validation of the glycolytic genes from FACS sorted macrophage population suggested no change in the genes associated with the glycolytic pathway (Fig. S4D). However, the macrophages in the cKO skin did not express GLUT1 (Fig. S4E). On the other hand, we observed and subsequently validated an increase in the expression of genes associated with the TCA cycle in FACS sorted F480+ve macrophages isolated from cKO skin compared to the controls (Fig. S4C,F). IF staining of F4/80+ve macrophages in the WT and itgβ1 cKO skin with TCA cycle enzymes—CS and IDH1 further corroborated the qPCR and NGS data (Fig. 4B,C). We next performed steady-state metabolomic analysis to quantify the levels of glycolysis and TCA cycle metabolites of the dermal compartment of cKO skin compared to WT. Metabolomic analysis suggested a decrease in glycolysis metabolites and an increase in levels of TCA cycle intermediate metabolites (Fig. 4D,E). Notably, we did not observe any change in the expression of enzymes associated with TCA cycle in FACS sorted CD45-ve fibroblasts (Fig. S4H). Instead, NGS analysis of FACS sorted CD45-ve fibroblasts showed increased upregulation of pathways associated with lipid, and not carbohydrate, metabolism (Fig. S4I). This suggested that macrophages in the dermal compartment might be the key drivers of the TCA cycle. Taken together, our analysis suggested that macrophages reduce dependence on glycolysis and instead augment the TCA cycle in the cKO skin. Interestingly, the metabolic states of the epidermis and macrophages are in clear contrast with each other which suggests "metabolic compartmentation" of glycolysis and TCA cycle between the epidermis and macrophages, respectively, in the inflamed cKO skin (Fig. 4F).

We reasoned that since macrophages are decreasing glucose uptake, they can utilize epidermally-derived lactate as an alternative fuel to drive the TCA cycle (Fig. S4G) (Hui et al, 2017; Faubert et al, 2017). IF staining analysis suggested increased membrane localization of lactate importer, MCT1, in the macrophage compartment (Fig. 4G(left),H). This suggested that macrophages can act as potential sinks for lactate which, in turn, can be utilized to drive TCA in the macrophage compartment. Notably, membrane expression of MCT1 correlated with increased generation of MMP9 and basement membrane disruption (LAM332) in the

cKO skin (Fig. 4G(middle, right),I,J). This suggested a possible role for epidermally-derived lactate in driving macrophage polarization during sterile inflammation in cKO skin. Taken together, these results led to the hypothesis that epidermally-derived lactate can be a potential source to drive macrophage effector state in the inflamed cKO skin (Fig. 4K).

## Preventing lactate transport in cKO skin inhibits pro-remodeling fate and TCA cycle enzyme expression in the macrophages

Previous studies have suggested a role for lactate in driving M2-like macrophage fate in inflammatory disease conditions (Noe et al, 2021; Zhang et al, 2020; Mu et al, 2018; Zhang et al, 2021). In previous studies, our work suggests macrophages in the cKO skin acquire an M2-like fate characterized by an increase in expression of MMP9 that in turn, leads to increased breakdown of the basement membrane (Bhattacharjee et al, 2021). We therefore sought to understand if lactate alone, in the absence of an inflammatory cue, drives M2-like macrophage response and increase MMP9 levels in the skin. To test this, we administered Na-Lactate (0.5 mg/mL) subcutaneously under the back skin of neonatal mice (P0 stage) and extracted the skin at post-natal P2 stage for analysis (Fig. 5A). The control group was treated with sterile-PBS/Saline. IF of back skin suggested a dramatic increase in the expression of MMP9 in the back skin of Na-Lactate treated mice compared to the control (Fig. 5B). This suggests that lactate alone is sufficient to increase MMP9 expression in mice skin. This potentially occurs by driving the TCA cycle in the skin (Fig. 5C).

Since both the epidermis and macrophage compartments in the itgβ1 cKO express lactate transporters, we reasoned that epidermally-derived lactate might be sufficient to drive macrophage metabolic program and in turn its effector function. To test our hypothesis, we treated pregnant dams harboring WT and cKO embryos with the MCT1 inhibitor, AZD3965 (Polanski et al, 2014) (Fig. S2D). Interestingly, inhibition of lactate uptake by dermal macrophages in itgβ1 cKO animals led to a significant decrease in the expression of MMP9 which further resulted in a remarkable reduction in ECM degradation compared to DMSO treated controls (Fig. 5D–F). Similarly, inhibition of MCT4 (expressed in epidermis) using the MCT1/4 blocker, syrosingopine (SYRO) (Benjamin et al, 2018) with the identical treatment paradigm resulted in a similar reduction in the MMP9 generation and ECM degradation compared to the controls (Fig. 5D–F). These results suggest that lactate uptake by macrophages is necessary for macrophages to acquire an M2-like pro-ECM remodeling state.

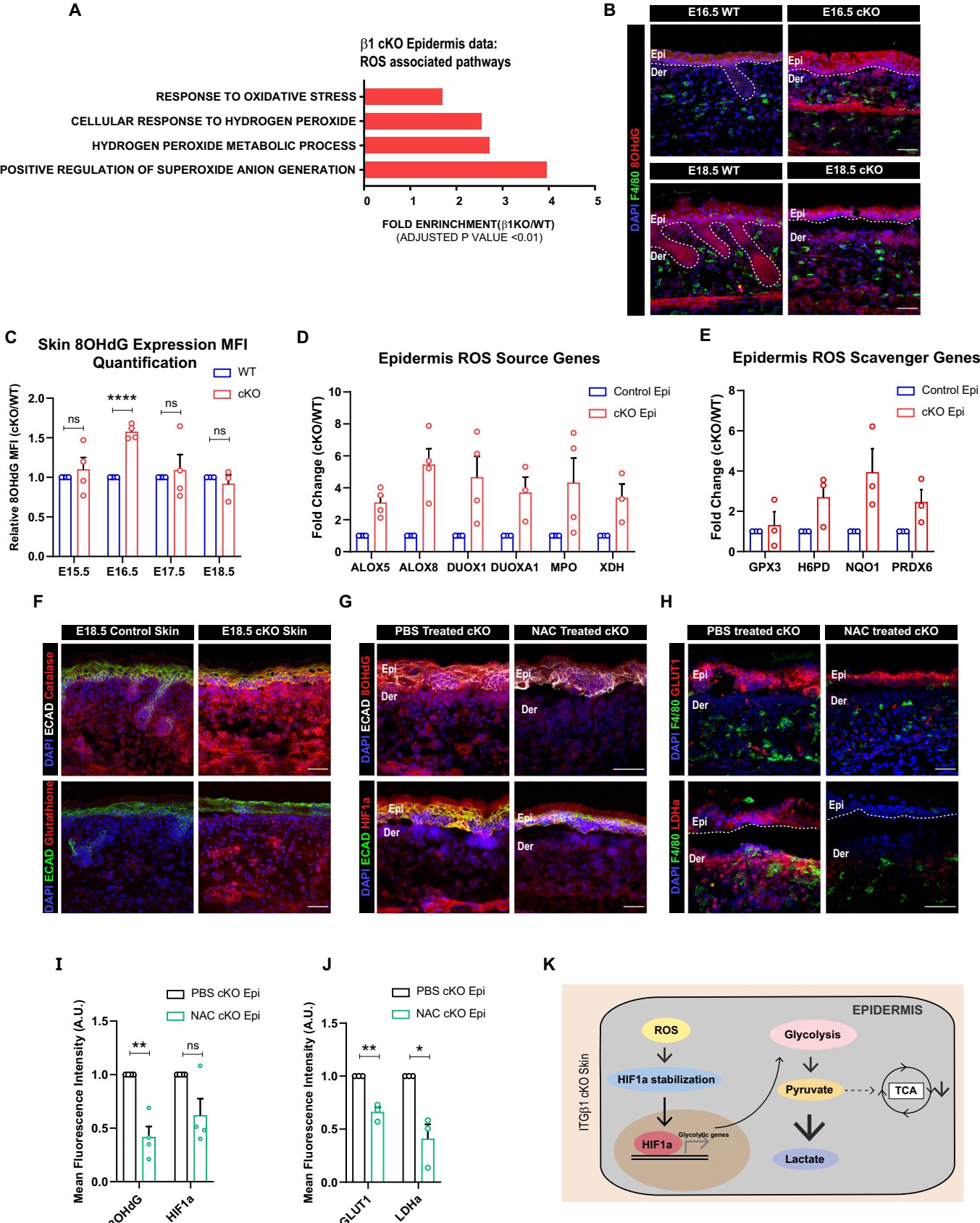

**Figure 3.  Increased ROS expression drives HIF1α induced glycolytic switch in epidermal compartment of itgβ1 cKO skin.**

(A) ROS-associated pathways upregulated in GSEA analysis of epidermal compartment in E18.5 cKO skin compared to controls (A). (B) IF immunostaining of cKO skin showing the expression of 8-OHdG (G) in E16.5 KO and control skin (N = 3). (C) Quantification of temporal change in the expression of 8-OHdG expression in cKO and control skin (N ≥ 3). (D) qPCR showing increased transcription of genes associated with ROS generation in the cKO epidermis compared to WT (N ≥ 3). (E) qPCR showing increased transcription of genes associated with ROS scavenging in the cKO epidermis compared to WT (N ≥ 3). (F) IF showing increased expression of Catalase and Glutathione in cKO skin compared to WT (N = 3). (G) cKO skin showing reduced expression of 8-OHdG and HIF1α after treatment with NAC compared to controls (N = 4). (H) cKO skin showing reduced expression of GLUT1 and LDHa quantified in (J) in NAC, treated cKO skin compared to controls (N = 3). (I) Quantification for (G) (N = 4). (J) Quantification for (H) (N = 3). (K) Schematic showing ROS-mediated regulation of HIF1α in cKO skin. Data Information: All the microscope images (B, F, G, H) were quantified (C, I, J) in ImageJ software. The number of biological replicates (N) for each panel is given in the figure legend. Scale bars: 50 µm. Student's t-test was performed for statistical analysis. *p ≤ 0.05, **p ≤ 0.01, ****p ≤ 0.0001, ns = not significant (student's t test). All graphs are presented as mean ± SEM. WT experimental values are normalized to 1 for relative quantification. Source data are available online for this figure.

To further test that epidermal specific lactate drives dermal macrophage responses, we injected SYRO in utero into the amniotic sac of E15.5 cKO embryos harbored in pregnant dams by performing survival surgeries. The control animals were treated with DMSO. We reasoned that in utero administration of SYRO would preferentially target the drug to the epidermal compartment of the embryonic skin as suggested in previous studies. The injected embryos were subsequently extracted and analyzed at embryonic day E18.5 (Fig. S7). Notably, inhibition of epidermal MCT4 using in utero SYRO injections lead to a remarkable reduction in the extent of basement membrane disruption and MMP9 expression (Fig. 5G,H). Taken together these results suggest that epidermal lactate is sufficient to drive M2-like fates in the dermal macrophages in cKO skin.

As shown above, M2-like pro-remodeling macrophages in the cKO skin are associated with increased TCA cycle enzyme expression. We therefore asked if inhibition of lactate transport between epidermis and macrophages using SYRO and AZD3965 would lead to reduced expression of TCA cycle enzymes in the macrophages. Notably, immunostaining analysis suggested a significant reduction in the expression of IDH1 in the macrophages in cKO skin treated with AZD3965 and SYRO compared to control (Fig. S5A,B). Interestingly, we did not observe a reduction in the expression of CS upon SYRO and AZD3965 treatment (Fig. S5A,C). This suggested that epidermally-derived lactate can drive M2-like polarization fate of dermal macrophages partly by driving the expression of IDH1 in cKO skin.

## Inhibition of ROS-HIF1α-glycolysis axis in epidermis and TCA cycle in macrophage attenuates M2-like macrophage fate in cKO skin

We next asked if augmentation of TCA cycle in the macrophages was necessary to drive their M2-like effector state in the cKO skin. To address this, we treated pregnant dams with the small-molecule inhibitors of the TCA cycle, pyromellitic acid (PA) (Beeckmans and Van Driessche, 1999), a fumarase inhibitor, and UK-5099 (Halestrap, 1975), a pyruvate uptake inhibitor (Fig. S2D). Interestingly, treatment with PA and UK5099 led to a significant reduction in the basement membrane disruption and MMP9 expression in cKO skin compared to the controls (Figs. 6A–C and S6A–C). These results suggested augmentation of TCA cycle in the macrophages is necessary to drive M2-like state in dermal macrophages in the cKO skin. These results additionally indicate that epidermally-derived lactate potentially drives the increased TCA cycle in the dermal macrophages in the cKO skin.

We next aimed to identify if epidermal metabolic state is coupled with the metabolic state and, in turn, the effector state of macrophages in the cKO skin. We, therefore, asked if inhibition of pathways upstream to lactate synthesis in the epidermis would lead to a reduction in the pro-remodeling fate and TCA cycle enzyme expression in the macrophages. To test this, we treated pregnant dams with glycolysis inhibitor, 2DG (2 deoxy D glucose) (Laszlo et al, 1960) using identical treatment paradigm (Fig. S2D). We reasoned that since lactate transport inhibition led to a reduction in the M2-like fate in the dermal macrophages, inhibition of lactate source i.e., glycolysis in the epidermal compartment, should have similar consequences. Remarkably, blocking glycolysis with 2DG led to a significant reduction in the generation of MMP9 and basement membrane disruption by macrophages (Fig. 6A–C). On the other hand, inhibition of fatty acid metabolism, which is primarily augmented by the fibroblast compartment, using etomoxir (Zhou and Grill, 1994) did not lead to reduction in macrophage-mediated ECM degradation (Fig. S4J). These results further suggested that the metabolic program augmented by the epidermal compartment, and not fibroblasts, have a direct consequence in regulating the effector functions of the dermal macrophages.

We next asked if inhibiting ROS and HIF1α, the upstream regulators of glycolysis, would lead to similar attenuation of macrophage pro-remodeling function in the cKO skin. As predicted, we observed a significant reduction expression of MMP9 and extent of ECM disruption in NAC and chetomin-treated cKO skin compared to the controls (Fig. 6D–H). Furthermore, as previously established that lactate potentially controls expression of TCA cycle enzyme IDH1 in the cKO skin, we asked if inhibition of glycolysis and its upstream regulators in the epidermis lead to a reduction in the expression of TCA cycle enzyme IDH1 in the cKO skin. Interestingly, 2DG, Chetomin and NAC treatment lead to a significant reduction in the expression of IDH1 in macrophages (Fig. S6D–G).

Overall, using small molecules to inhibit metabolic pathways, we show that perturbing lactate synthesis by attenuating ROS, HIF1α, and glycolysis in the epidermal compartment leads to the attenuation of M2-like pro-ECM remodeling fate in dermal macrophages in the cKO skin. This is partly linked to the consequent reduction in the expression of TCA cycle enzymes in the dermal macrophages.

## Epidermis-derived lactate reprograms macrophage metabolism and effector function through NF-κB activation

We next aimed to develop a mechanistic understanding of how lactate controls the expression of MMP9 and TCA cycle genes in

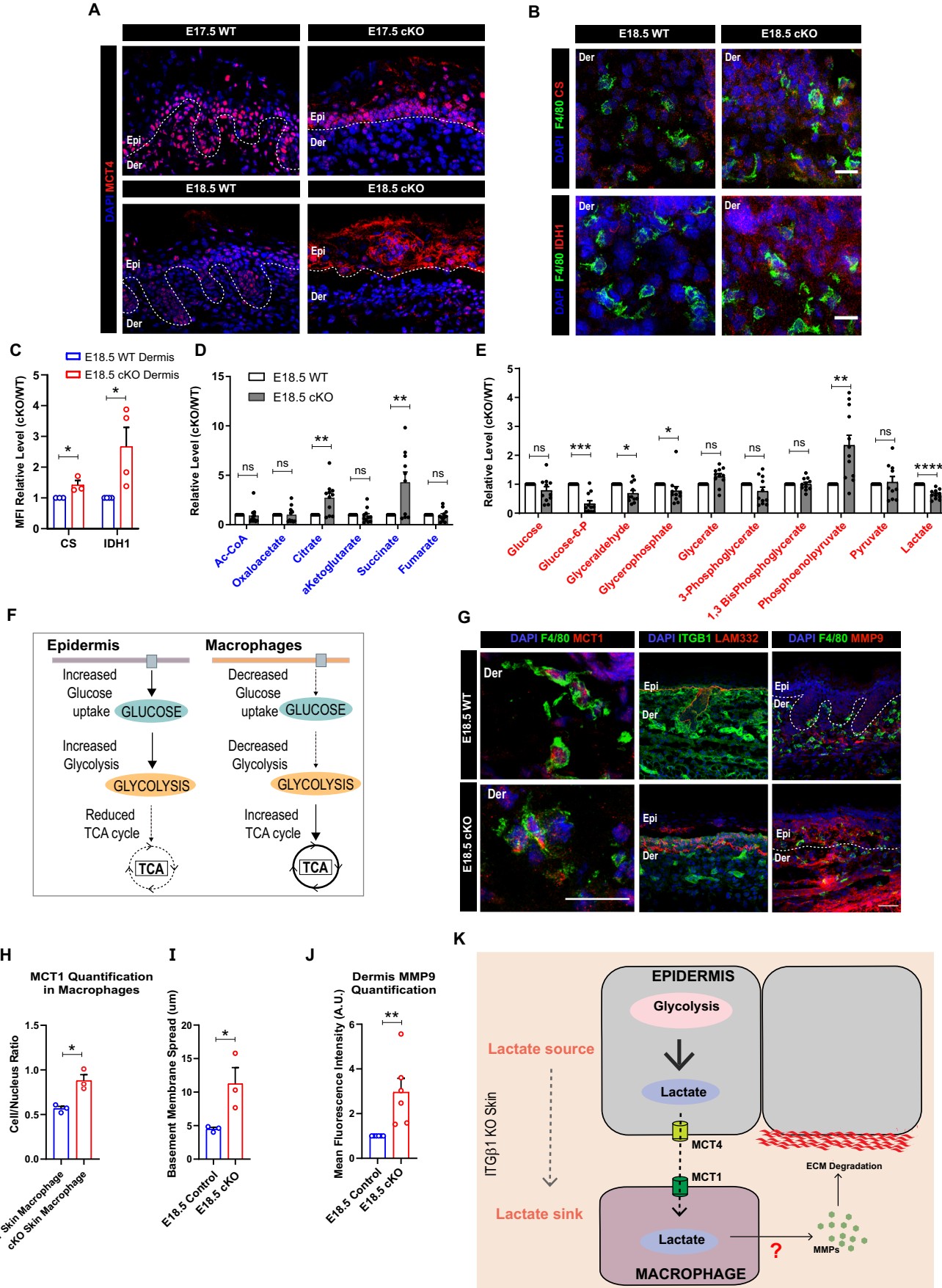

**Figure 4.   Epidermis is an exporter while macrophages are importers of lactate metabolite in epidermal itgβ1 cKO skin.**

(A) Increased membrane expression of lactate exporter, MCT4, in epidermal compartment at embryonic stage E17.5 and E18.5 cKO skin compared to controls (A) (N = 3). (B, C) Increased expression of TCA cycle enzymes CS (N = 3) and IDH1 (N = 4) (B), quantified in (C) in macrophage compartment in cKO skin compared to controls. (D, E) Relative levels of TCA cycle metabolites (D) and glycolytic metabolites (E) quantified using steady-state metabolomics in the dermal compartment in E18.5 cKO skin compared to WT (N = 5). (F) Schematic representing compartment separation of Glycolysis and TCA cycle in epidermal and macrophages compartment in cKO skin compared to controls. (G) (left) Increased membrane expression of lactate importer MCT1 (G, left) in macrophages in the cKO skin compared to controls (representative for N = 3, n = 30–50 cells each). Scale bars: 20 µm. (G) (middle, right) Increased basement membrane (LAM332) disruption in cKO skin compared to WT (G, middle). Increased in expression on MMP9 in the cKO skin compared to WT (G, right). (H) Quantification of membrane expression (MCT1 Cytoplasmic/Nuclear localization) of MCT1 protein in the macrophages in E18.5 control and cKO skin. (I, J) Quantification of basement membrane (LAM332) spread (I) and MMP9 expression (J) in cKO skin compared to WT (N = 3). (K) Schematic representing epidermis as a potential source and macrophages as potential sink for lactate metabolite in cKO skin. Acquisition of lactate by macrophage compartment correlates with increased generation of matrix remodeling enzymes that result in ECM degradation. Data Information: All the microscope images (A, B, G) were quantified (C, H–J) in ImageJ software. The number of biological replicates (N) for each panel is given in the figure legend. n in figure (G, left) represents the number of macrophages. Scale bars: 50 µm (Except panel (G, left)). Student's t-test was performed for statistical analysis. *$p \le 0.05$, **$p \le 0.01$, ***$p \le 0.001$, ****$p \le 0.0001$, ns=not significant (student's t test). All graphs are presented as mean ± SEM. WT experimental values are normalized to 1 for relative quantification. Source data are available online for this figure.

the macrophages in the cKO skin. Interestingly, several studies have implicated a role for lactate as a signaling molecule, capable of inducing NF-κB signaling in macrophages (Samuvel et al, 2009; Végran et al, 2011). In addition, NF-κB has been shown to be a transcriptional regulator of MMP9 and IDH1 enzyme of the TCA cycle (Zhou et al, 2017). Moreover, we previously reported an increase in the expression of p65 NF-κB (active) in the cKO skin (Kurbet et al, 2016). Taken together, we hypothesized that epidermis-derived lactate can direct MMP9 synthesis and TCA cycle augmentation through NF-κB signaling. We first asked if the inhibition of lactate transport between epidermis and macrophages using SYRO and AZD3965 leads to a reduction in p65 NF-κB expression in the macrophages. Notably, we observed a significant reduction in the expression of NF-κB (active) in macrophages of SYRO and AZD3965-treated cKO skin compared to the controls (Fig. 7A,C). In addition, inhibition of glycolysis and HIF1α using Chetomin also led to inhibition of p65 NF-κB in the macrophage of cKO skin compared to controls (Fig. 7B,D). Altogether, these results suggested that NF-κB in macrophages is a downstream target of epidermis-derived lactate in the cKO skin.

To understand if NF-κB controls MMP9 synthesis and expression of IDH1 in macrophages, we inhibited NF-κB specifically in macrophages using Bay-11-7082, an NF-κB signaling inhibitor (Rauert-Wunderlich et al, 2013; Mori et al, 2002). To specifically target BAY-11-7082 to macrophages, we administered slow release liposomally-encapsulated BAY-11-7082 in utero to embryos at E15.5 and recovered animals at E18.5 for analysis (Fig. S7). As expected, BAY-11-7082 treatment led to a significant reduction in the expression of p65 NF-κB in the cKO skin compared to the controls (Fig. 7E,F). Strikingly, BAY-11-7082 mediated reduction in NF-κB signaling led to a significant reduction in the expression of MMP9 (Fig. 7E,G). These results suggest that epidermis-derived lactate drives M2-like pro-remodeling state in dermal macrophages partially by activating NF-κB signaling.

Taken together, our results indicate that inflammation in the cKO skin is associated with significant changes in the metabolic states of the epidermis and macrophages. This metabolic reprogramming is initiated due to an early ROS-HIF1α expression in the epidermis that leads to enhanced glycolysis that, in turn, causes exacerbated synthesis of lactate metabolite in the epidermis. The lactate released from the epidermal compartment reprograms macrophage metabolism and, in turn, its functional state through

activation of NF-κB. NF-κB signaling in macrophages causes enhanced transcription of TCA cycle enzymes (IDH1) and ECM-remodeling enzymes (MMP9) that causes exacerbated degradation of the basement membrane between the epidermis and dermis. Enhanced ECM degradation of the basement membrane causes increase in skin inflammation through generation of DAMPs (Fig. 7H).

## Inhibition of lactate transport using Syrosingopine inhibits psoriasis in imiquimod-treated mice model

Lastly, we aimed to understand the therapeutic applicability of our findings in the cKO skin. Previous studies on imiquimod induced mice model of psoriasis suggested increase in glucose utilization, glycolytic intermediates, and lactate in the epidermal compartment (Zhang et al, 2018). We therefore hypothesized that lactate generated as a result of increased glycolysis in psoriatic epidermis might potentially drive psoriatic skin disease by augmenting fate changes in the macrophage compartment. Analysis of the mice skin post 5 days of imiquimod treatment suggested increased epidermal hyperproliferation and thickening which was concomitant with increased monocyte and macrophage burden (Fig. S8A–C). In addition, we further observed increased MMP9 expression in the dermal compartment (Fig. S8D). Interestingly, increased epidermal hyperproliferation in the imiquimod induced psoriatic mice skin was associated with increased expression of GLUT1 and MCT4 in the epidermis compared to Vaseline treated control (Fig. 8A,C). This suggested that the epidermal compartment in the imiquimod induced mouse model of psoriasis could potentially export lactate. In addition, macrophages show increased expression of TCA cycle enzymes, CS and IDH1, lactate importer MCT1, and p65 NF-κB (Fig. 8B,D). This suggested that macrophages in the psoriatic skin potentially import lactate to drive TCA cycle which, in turn is necessary for their pro-remodeling fate switch. We finally asked if inhibition of lactate transport between epidermis and macrophages prevents macrophage polarization and in turn, psoriasis development in imiquimod induced mice model of psoriasis. To test this, we started treating imiquimod mice with intraperitoneal doses of SYRO (10 mg/kg) from day 3 onwards (Fig. S8E). The animals were sacrificed on day 6 and back skin samples were collected for analysis (Fig. S8E). Significantly, treatment of mice with Syrosingopine led to a dramatic reduction in epidermal plaques and epidermal hyperproliferation, monocyte-macrophage burden, and

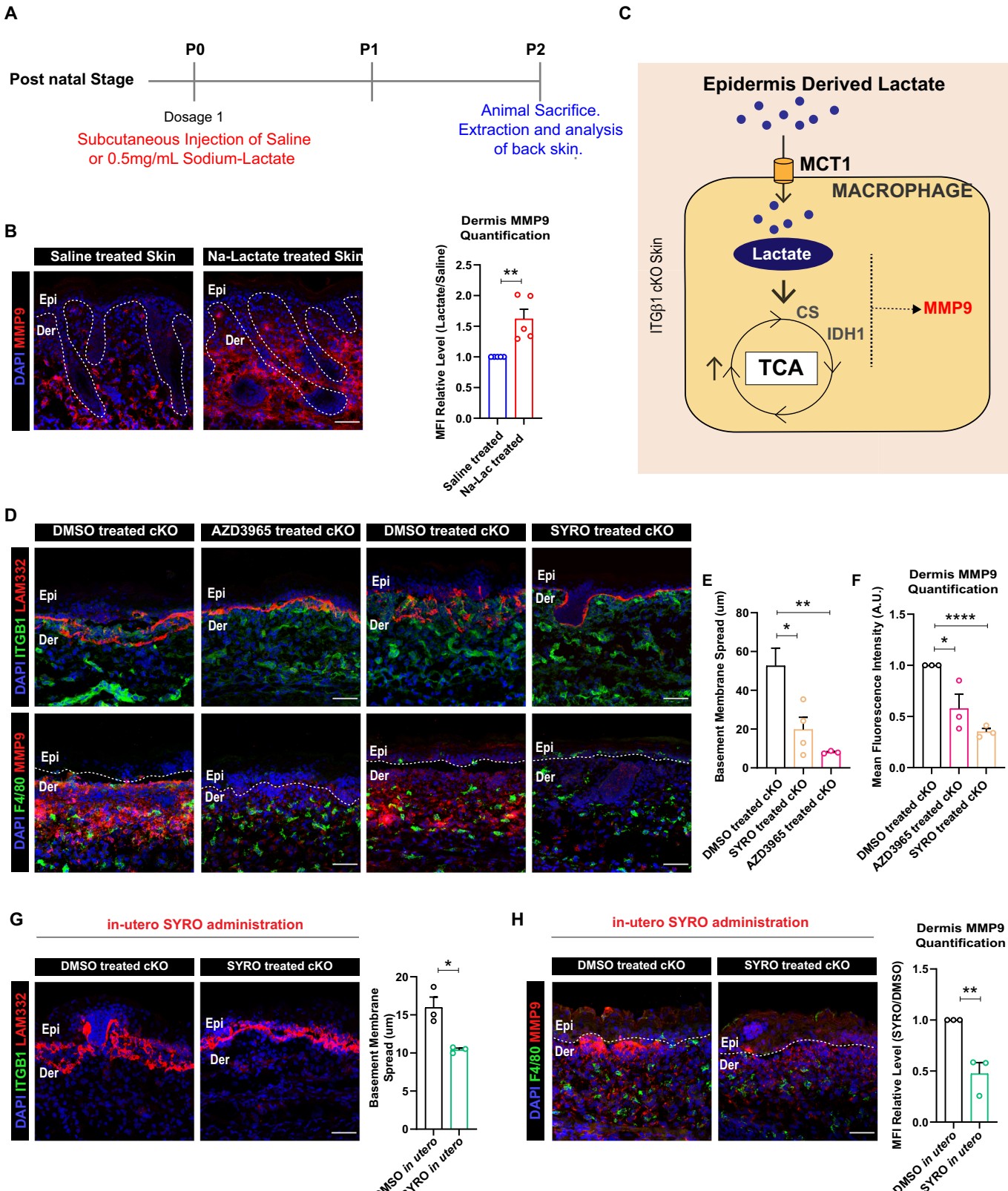

**Figure 5. Inhibition of lactate transport between epidermis and macrophages reduces ECM degradation by macrophages in cKO skin.**

(A) Schematic showing dose schedule for Sodium-lactate administered subcutaneously in the back skin of neonatal mice. (B) P2 stage back skin showing increased expression of MMP9 after subcutaneous administration of Na-Lactate at P0 stage compared to controls ($N = 5$). (C) Schematic depicting the hypothesis that epidermally-derived lactate drives TCA cycle and regulated MMP9 expression in dermal macrophages in the cKO skin compared to WT. (D–F) Decreased MMP9 expression and basement membrane (LAM332) remodeling in cKO skin treated with MCT1 inhibitor, AZD3965, and MCT4/MCT1 inhibitor, Syrosingopine compared to DMSO treated control cKO skin (D). Basement membrane spread quantified in (E) and MMP9 levels in (F) ($N = 3$). (G, H) cKO skin showing reduced degradation of LAM332 (G) and expression of MMP9 (H) after in utero administration of Syrosingopine compared to DMSO administered controls ($N = 3$). Data Information: All the microscope images (B, D, G, H) were quantified (B, E–H) in ImageJ software. The number of biological replicates ($N$) for each panel is given in the figure legend. Scale bars: 50 µm. Student's t-test was performed for statistical analysis. *$p \leq 0.05$, **$p \leq 0.01$, ****$p \leq 0.0001$ (student's t test). All graphs are presented as mean ± SEM. WT experimental values are normalized to 1 for relative quantification. Source data are available online for this figure.

MMP9 and p65 NF-κB expression in the psoriatic skin compared to the controls (Figs. 8E–H and S8F). We additionally observed a remarkable reduction in the transcription of psoriatic biomarkers—*Il6*, *Il17a*, *Il23a*, and *Tnfa* (Fig. 8I). These results establish that in imiquimod induced mice psoriatic skin, epidermis-derived lactate exacerbates psoriatic burden by driving MMP synthesis by macrophages through activation of NF-κB. We also establish that treatment with SYRO attenuates psoriasis in imiquimod induced mice model of psoriasis partly by abrogating the lactate-NF-κB-MMP9 axis (Fig. 8J).

## Discussion

In the skin, how metabolism controls cell fates during inflammatory skin disease conditions is only beginning to be understood (Cibrian et al, 2020). Much of the ongoing work on metabolism in skin has been focused on understanding metabolic reprograming in the epidermal compartment of the skin. In this regard, recent studies suggest that the epidermis increases glucose uptake in inflammatory skin disease conditions such as psoriasis and atopic dermatitis (Zhang et al, 2018; Choi et al, 2020). Prevention of glucose uptake in the epidermal compartment led to a remarkable reduction in the innate and adaptive immune cell pool in the dermis (Zhang et al, 2018). While such studies only address the metabolic changes associated with a single compartment, how different compartments in the skin communicate through metabolites and partition local nutrient resources to drive their functional states remains poorly understood. As inflammation is an orchestrated phenomenon comprising of several different cell types acting in concert, understanding metabolic crosstalk between these cell types will aid in developing a deeper understanding of the molecular events involved in the initiation and progression of disease.

Interestingly, during inflammation, our work shows compartmentation of glycolysis and TCA cycle in the epidermis and macrophages, respectively. While there are several examples of inter-organ metabolic compartmentation in maintaining systemic metabolic homeostasis, metabolic compartmentation of the respiratory carbon chain during inflammation within tissues remain to be understood. Metabolic compartmentation not only imparts cell types with unique metabolic states that supports and directs their effector function, but also promotes efficient resource partitioning (Bar-Peled and Kory, 2022). Notably, certain reports on cancer have indicated the possibility of such metabolic compartmentation where lactate derived from 'glycolytic' cancer cells acts as a carbon source for 'non-glycolytic' cells in the cancer niche—including the immune cells and cancer-associated fibroblasts (Hui et al, 2017; Faubert et al, 2017). This suggests that the findings obtained from our study can be extrapolated to understand metabolic compartmentation in different inflammatory disorders such as cancer.

In addition to cancers, our results can be extrapolated to understand macrophage metabolism in several auto-immune disorders such as rheumatoid arthritis (RA). Previous studies on RA suggest that increased lactate levels at the site of inflammation augments a pro-inflammatory phenotype in helper-T cells (Pucino et al, 2019; Fujii et al, 2015). In addition to T cell population, RA is also associated with increase in local macrophage population that exacerbate disease symptoms (Cutolo et al, 2022). We speculate that similar to our findings, increase local lactate levels can potentially polarizes macrophages into an M2-like fate which further exacerbates local RA-associated inflammation by degrading ECM through excess MMPs synthesis.

Conventionally, inflammatory skin diseases have been thought to be primarily driven by inflammatory cytokines. We, and others, have previously shown that the epidermal compartment of the skin, is the major source of cytokines and chemokines that leads to increased infiltration and pro-remodeling fate acquisition in macrophages (Bhattacharjee et al, 2021; Borgia et al, 2021). Consequently, several skin diseases are currently treated by biologics against inflammatory cytokines such as IL-17, TNF-α and IL-23 (Di Meglio et al, 2014; Rendon and Schäkel, 2019). However, these biologics lead to global immunosuppression which lead to several co-morbidities (Yamazaki, 2021). Hence there is need to identify unique molecular targets to improve therapeutic outcomes from severe inflammatory skin disorders. Our study suggests that targeting metabolism at the level of crosstalk between disease-associated cell types in the skin can be a novel therapeutic avenue that can be explored for treating inflammatory skin disorders.

Psoriasis and atopic dermatitis have been shown to be associated with increased systemic levels of lactate which may contribute to systemic inflammation (Yan, 2017). Importantly, systemic level of inflammation associated with skin disease conditions contribute to additional musculoskeletal and cardiovascular co-morbidities (Boehncke, 2018; Verma et al, 2021). While the increased glycolysis in the activated T cells and macrophages have been suggested to be the source of lactate, our results suggest that the epidermal compartment could be the primary source of enhanced systemic lactate level in these disease conditions. The enhanced circulating levels of lactate might be a source of systemic inflammation. This suggests that targeting lactate metabolism might also aid in reducing the systemic co-morbidities associated with inflammatory skin diseases such as psoriasis and atopic dermatitis.

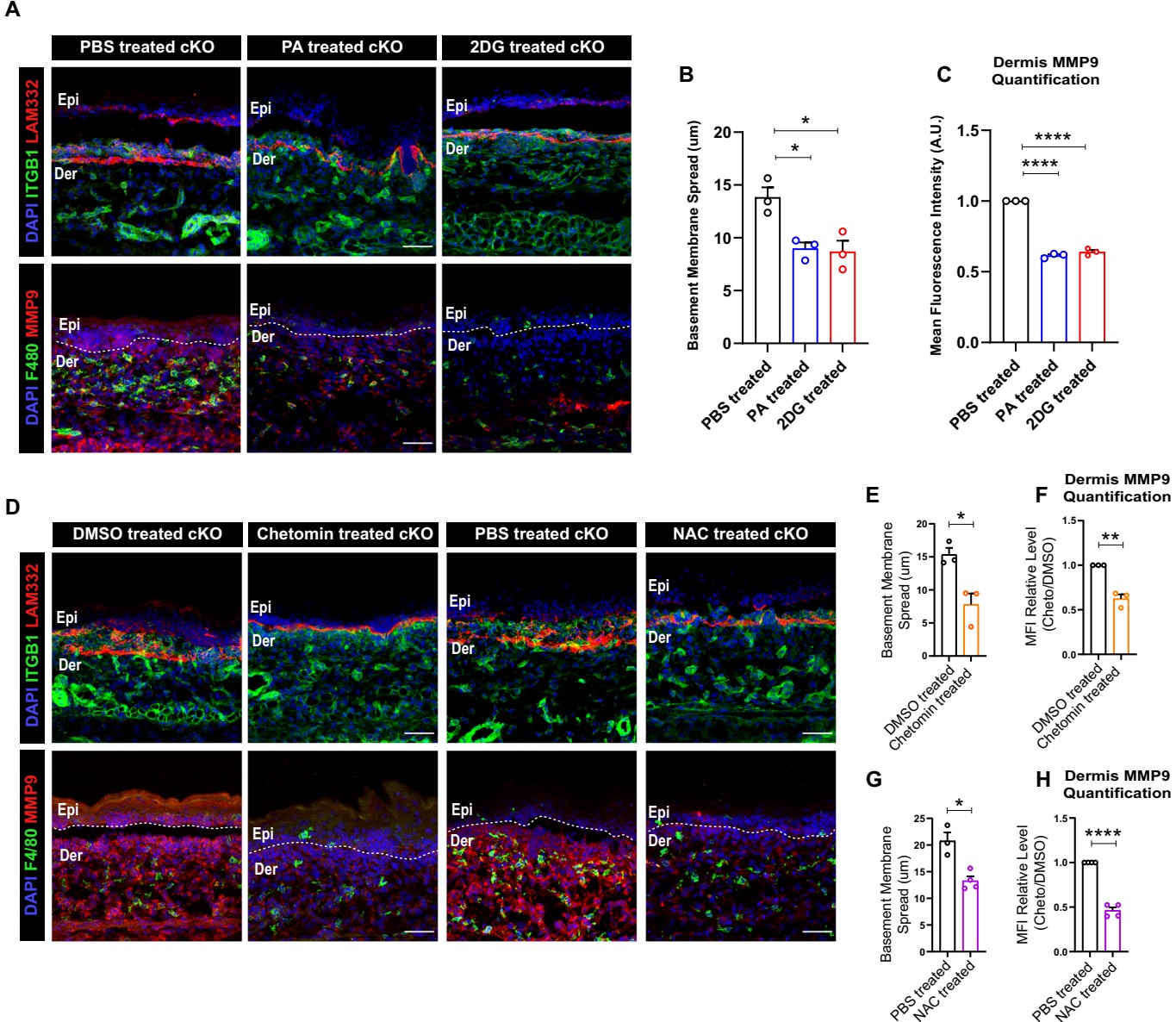

**Figure 6. Inhibition of ROS and HIF1α inhibits basement membrane remodeling and TCA cycle enzyme, IDH1 expression in cKO skin.**

(A–C) Decreased basement membrane disruption (top) and MMP9 expression (bottom) in cKO skin treated with TCA cycle inhibitor, PA, and Glycolysis inhibitor, 2DG (A). Quantification of basement membrane spread (B) and MMP9 expression (C) in cKO skin treated with TCA cycle inhibitor, PA, and Glycolysis inhibitor, 2DG ($N = 3$). (D–H) Decreased MMP9 expression and basement membrane (LAM332) remodeling in cKO skin treated with HIF1α inhibitor, Chetomin and ROS inhibitor, NAC compared to controls (D) (representative for $N = 3$). Quantification of Basement membrane spread and MMP9 expression in cKO skin treated with HIF1α inhibitor, Chetomin and ROS inhibitor, NAC compared to controls (E–H) ($N = 3$). Data Information: All the microscope images (A, D) were quantified (B, C, E–H) in ImageJ software. The number of biological replicates ($N$) for each panel is given in the figure legend. Scale bars: 50 μm. Student's t-test was performed for statistical analysis. *$p \leq 0.05$, **$p \leq 0.01$, ****$p \leq 0.0001$ (student's t test). All graphs are presented as mean ± SEM. WT experimental values are normalized to 1 for relative quantification. Source data are available online for this figure.

In the study, we propose that lactate drives macrophage effector function though NK-κB activation. While the NF-κB activation alludes to the signaling roles of lactate metabolite, we believe that there might be additional mechanisms through which lactate might polarize macrophages in the inflamed skin. Interestingly, extracellular lactate can act as source of carbons for generating acetyl-CoA that changes acetylation states in tumor-associated macrophages (Noe et al, 2021). In addition, lactate can remodel

chromatin activation state through lactylation or be directly fed into the TCA cycle metabolites which can also be used as epigenetic adducts which can cause major changes in the transcriptional output of the immune cells (Martínez-Reyes and Chandel, 2020). These epigenetic modifications might influence the transcriptional landscape in the dermal macrophages that might explain the changes observed in the TCA cycle enzymes in the macrophage compartment. We speculate that lactate and TCA cycle

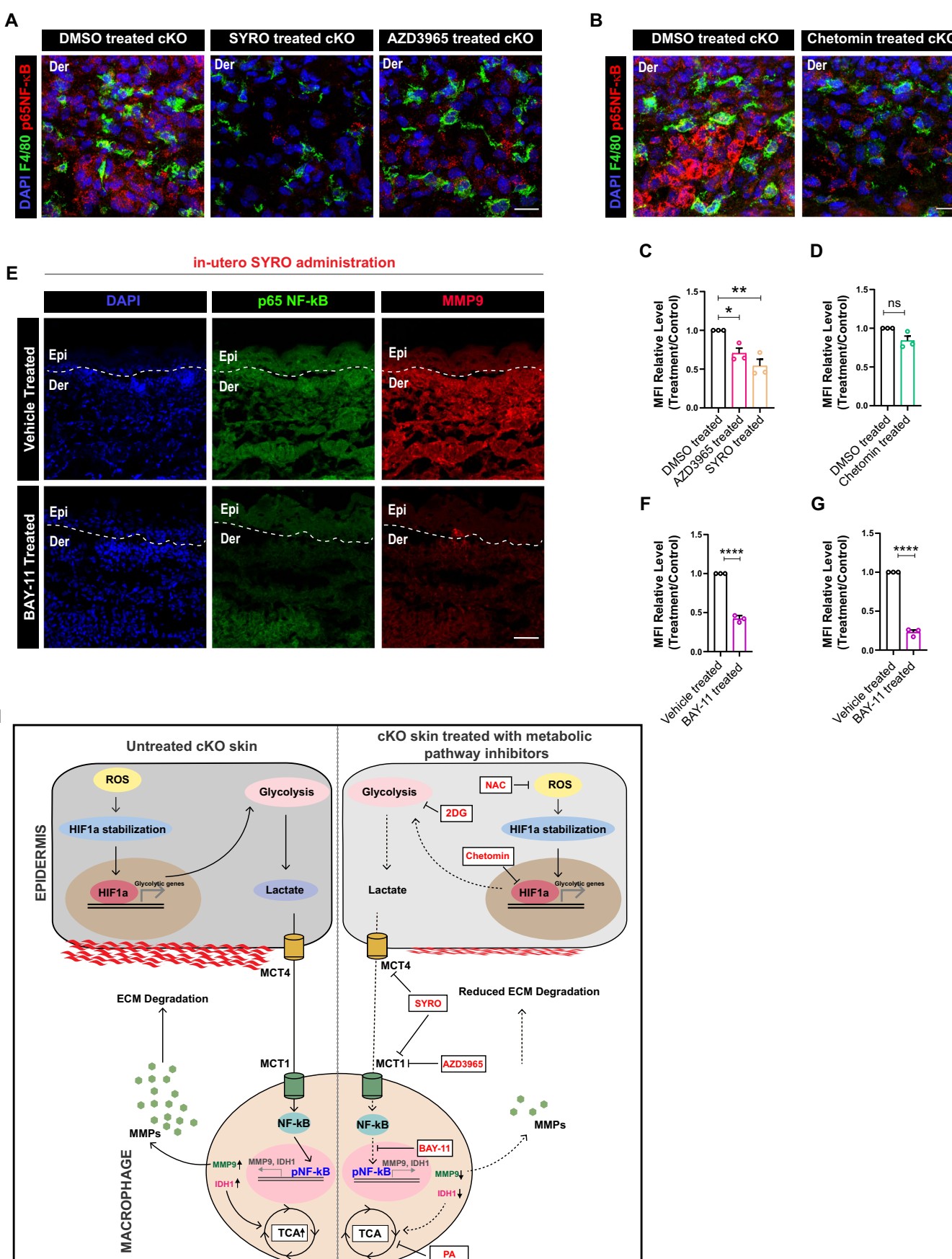

◀ **Figure 7. Lactate augments MMP9 and IDH1 expression in dermal macrophages through NF-κB activation in cKO skin.**

(A) Macrophages in the cKO skin showing decreased expression of p65 NF-κB after AZD3965 and SYRO treatment compared to controls (A) (N = 3). (B) Macrophages in the cKO skin showing decreased expression of p65 NF-κB after chetomin treatment compared to controls (N = 3). (C) Quantification for (A) (N = 3). (D) Quantification for (B) (N = 3). (E–G) cKO skin showing reduced expression of p65 NF-κB (E, top), quantified in (F), and MMP9 (E, bottom), quantified in (G) (N = 3) after in utero administration of liposome-encapsulated Bay-11-7082 in the amniotic sac compared to controls. (H) Summary schematic proposing that under sterile inflammatory conditions in skin, early augmentation of ROS-HIF1α axis leads to enhanced glycolysis and lactate generation in the epidermal compartment. Macrophages, in turn, uptake lactate through MCT1 thereby acting as a sink for epidermally-derived lactate. The lactate subsequently augments pro-remodeling fate switch in macrophages through activation of NF-κB signaling, that in turn, increases expression of TCA cycle enzymes and MMP9 in the macrophages. Consistently, inhibition of lactate transporters, epidermal-macrophage intrinsic metabolism, and upstream glycolysis regulators in the epidermal compartment using small-molecule inhibitors lead to inhibition of pro-remodeling fate and hence inflammation in the cKO skin. Data Information: All the microscope images (A, B, D) were quantified (C, D, F, G) in ImageJ software. The number of biological replicates (N) for each panel is given in the figure legend. Scale bars: 50 μm. Student's t-test was performed for statistical analysis. *$p \leq 0.05$, **$p \leq 0.01$, ****$p \leq 0.0001$ (student's t test). All graphs are presented as mean ± SEM. WT experimental values are normalized to 1 for relative quantification. Source data are available online for this figure.

intermediates can additionally contribute to changes in the epigenetic landscape of macrophages that, in turn, may lead to enhanced MMP synthesis. This hypothesis needs further investigation.

To conclude, our study identifies a lactate-mediated crosstalk that drives sterile inflammation and psoriatic skin disease. The ability of lactate transport inhibitors to block the progression of psoriasis in our mouse models provides an exciting avenue to identify additional "druggable" metabolic pathways to treat inflammatory skin diseases.

# Methods

## Animal study

Integrin β1 cKO animals (mixed background) were generated by crossing ITGβ1fl/+|KRT14-Cre males with ITG β1fl/fl (C57B6J background) females. ITG β1fl/+|KRT14-Cre males were generated by crossing KRT14-Cre homozygous males (CD1 background) with ITG β1fl/fl (C57B6J background). Since integrin β1 cKO embryos are neonatally lethal, all the experiments reported in the current study have been done on embryos extracted from euthanized dams at specific embryonic stages.

Pregnant dams containing the cKO and the WT embryos were housed at NCBS/inStem ACRC (Animal Care and Resource Centre) facility. Handling, breeding, and euthanization of animals were done in accordance with the guidelines and procedures approved by the inStem IACUC (Institutional Animal Care and Use Committee). All experimental and breeder cages were maintained in SPF2 (Specific pathogen-free 2) facility with standard ventilation, temperature (21 °C), 12-h light and dark cycle, and sterilized food and water.

## Drug treatments for β1 cKO animals

The pregnant dams containing the WT and integrin β1 cKO embryos were treated with small-molecule inhibitors of specific metabolic pathways. All animals were treated for 3 days starting from E15.5. Embryos were extracted on E18.5 and analyzed. In control experiments, pregnant dams were treated with the vehicles such as sterile PBS or 5% DMSO.

The liposome-mediated inhibition of NF-κB pathway using BAY-11-7082 has been described previously (Kurbet et al, 2016).

In utero injection of lactate transport inhibitor Syrosingopine was done as described previously (Beronja and Fuchs, 2013).

The details of the drugs and treatment schedule is given in Table 1.

## Imiquimod induced mice model for psoriasis

C57B6/J mice back skin was shaved and 125 mg of commercially available 5% imiquimod (Glenmark) was applied topically daily on the shaved back skin. Vaseline was used as control for the above experiment. After 5 days of daily imiquimod or Vaseline application, animals were euthanized as per guidelines and procedures approved by the inStem IACUC (Institutional Animal Care and Use Committee). The back skin was collected for further analysis. All experimental and breeder cages were maintained in SPF2 (Specific pathogen-free 2) facility with standard ventilation, temperature (21 °C), 12-h light and dark cycle, and sterilized food and water.

## Drug treatments for imiquimod animal models

For lactate transport inhibition experiments, 125 mg of imiquimod (Glenmark) was applied on the back skin of the mice induce psoriasis for a total of 5 days. From the third day onwards, animals were treated with intraperitoneal doses of Syrosingopine (SML-1908, SIGMA) at 10 mg/kg concentration. The control animals were treated with 5% DMSO in sterile 1XPBS. Both male and female C57B6/J were used in these experiments. After 5 days, mice were euthanized as per guidelines and procedures approved by the inStem IACUC (Institutional Animal Care and Use Committee). The back skin was collected for further analysis. All experimental and breeder cages were maintained in SPF2 (Specific pathogen-free 2) facility with standard ventilation, temperature (21 °C), 12-h light and dark cycle, and sterilized food and water.

## Sodium-lactate injections in neonatal mice pups

0.5 mg/mL concentration of sodium lactate (SIGMA, gifted by Dr. Sunil Laxman Lab, InStem) was prepared in sterile 1XPBS solution. Approximately 25 μl of the solution was injected in the upper back skin of neonatal animals using an insulin syringe. Sterile 1XPBS was injected as a control experiment. The animals were rested for single day and sacrificed on post natal day P2. This was followed by extraction of the back skin and analysis of MMP9 expression using IF.

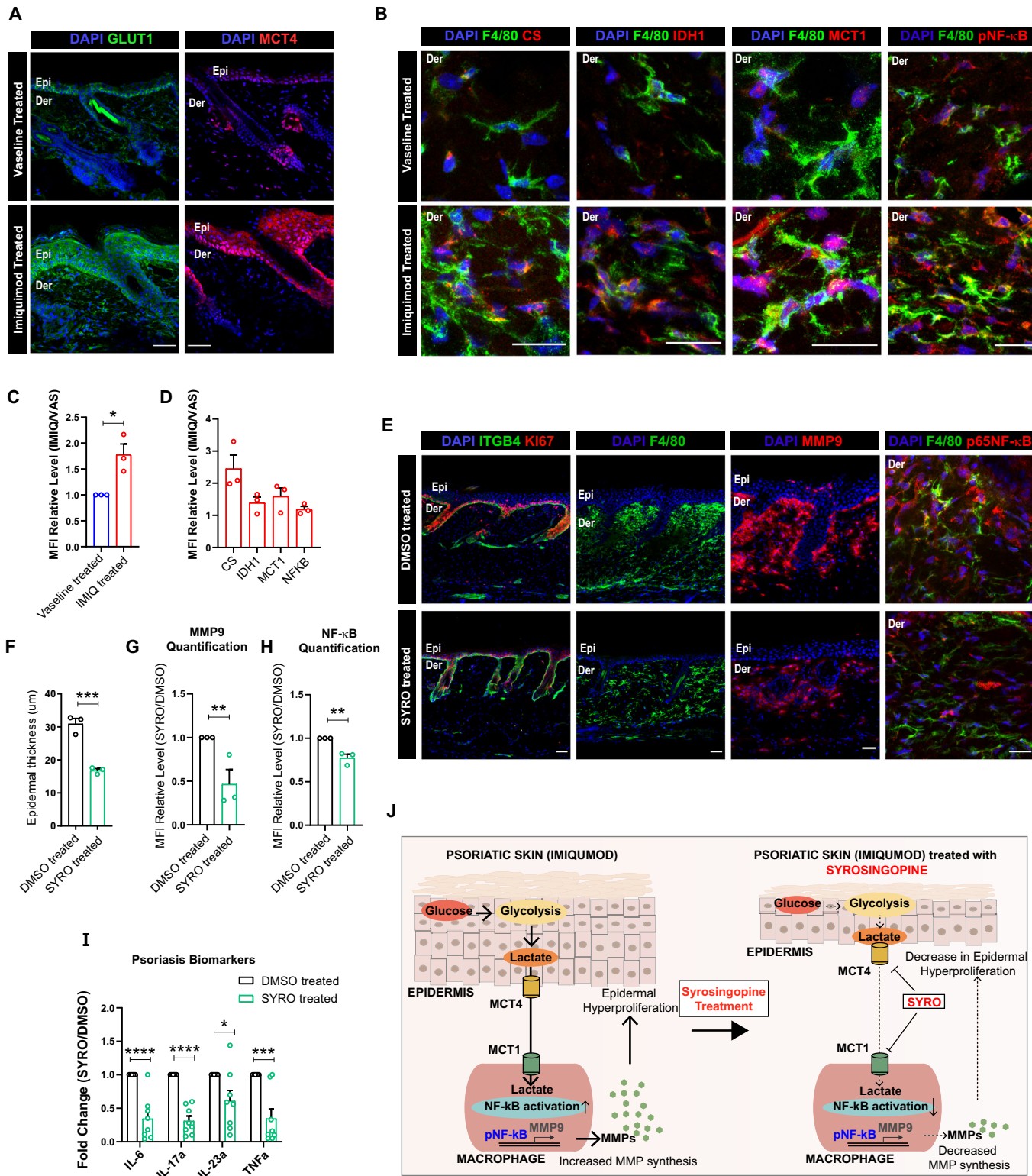

## Immunofluorescence staining

Embryos extracted from euthanized pregnant dams were frozen in tissue freezing media (OCT) and 10-micron cryosections were collected on charged glass slides and stored in −80 °C. For immunostaining, cryosections were thawed in room temperature (RT) for 5 min and fixed in acetone (Merck) for 5 min at −20 °C or 4% paraformaldehyde (Sigma) at room temperature for 10 min. Paraformaldehyde fixed sections were permeabilized using permeabilization solution—1XPBS plus 0.2–0.5% Triton X-100

**Figure 8. Treatment with Syrosingopine attenuates psoriasis symptoms in imiquimod-treated mice.**

(A) Adult (2 months) mice skin showing increased epidermal expression of GLUT1 and MCT4 (A), quantified in (C), after 5 day imiquimod treatment in skin compared to controls (N = 3). (B) Macrophages (F4/80, green) showing increased expression of CS, IDH1, MCT1, and p65 NF-κB (from left to right), quantified in (D), in imiquimod treated skin compared to controls (N = 3). (C) Quantification for GLUT1 expression in (A) (N = 3). (D) Quantification for (B) (N = 3). (E–H) Reduction of epidermal thickening and number of proliferative cells (Ki67, red), macrophage burden, MMP9 expression, and NF-κB expression (E, from left to right) in Syrosingopine treated C57B6/J mice compared to 5% DMSO treated control in imiquimod induced mice model of psoriasis (N = 3). Quantification of epidermal thickness (F) (N = 3), MMP9 levels (G) (N = 3), and NF-κB expression in macrophages (H) after Syrosingopine administration in imiquimod-treated mice compared to control (N = 3). (I) qPCR showing downregulation of Psoriatic biomarkers in the imiquimod induced psoriatic mice whole skin after treatment with Syrosingopine compared to control (N = 8). (J) Schematic showing that epidermally-derived lactate drives psoriasis through enhanced NF-κB activation in the macrophages that, in turn, leads to enhanced generation of MMP9. Inhibition of lactate-mediated crosstalk between epidermis and macrophages using Syrosingopine attenuates psoriatic burden in imiquimod-treated mice. Data Information: All the microscope images (A, B, E) were quantified (C, D, F–H) in ImageJ software. The number of biological replicates (N) for each panel is given in the figure legend. Scale bars: 50 µm. Student's t-test was performed for statistical analysis. *$p \leq 0.05$, **$p \leq 0.01$, ***$p \leq 0.001$, ****$p \leq 0.0001$ (student's t test). All graphs are presented as mean ± SEM. WT experimental values are normalized to 1 for relative quantification. Source data are available online for this figure.

**Table 1. Table showing details of all drugs used for the manuscript.**

| S.No. | Drug | Company | Catalog number | Target | Dosage | Dosage Route |
|---|---|---|---|---|---|---|
| 1. | 2DG (2 Deoxy D Glucose) | Sigma | D8375 | Glycolysis | 1 g/kg concentration | Intraperitoneal |
| 2. | AZD3975 | MedChem | HY-12750 | MCT1 – Lactate Uptake Inhibitor | 200 mg/kg concentration | Intraperitoneal |
| 3. | Syrosingopine | Sigma | SML1908 | MCT1/MCT4 – Lactate transport inhibitor | 10 mg/kg concentration | Intraperitoneal and Intrauterine |
| 4. | UK-5099 | Sigma | PZ0160-25MG | Mitochondrial Pyruvate carrier 1 inhibitor | 10 mg/kg concentration | Intraperitoneal |
| 5. | NAC | Sigma | A7250-10G | Reactive oxygen species inhibitor | 200 mg/kg concentration | Intraperitoneal |
| 6. | PA | Sigma | B4007 | Fumarase Inhibitor | 100 mg/kg concentration | Intraperitoneal |
| 7. | Chetomin | Sigma | C9623 | HIF1α dependent transcription inhibitor | 3 mg/kg concentration | Intraperitoneal |
| 8. | Etomoxir | Sigma | E1905 | CPT1a (Fatty acid oxidation) inhibitor | 100 mg/kg concentration | Intraperitoneal |
| 9. | Bay-11-7082 | Sigma | B5556 | NF-κB signaling inhibitor | 3 mg/kg concentration | Intrauterine |

(Sigma) for 10 min at RT. Fixed and permeabilized sections were blocked using 5%NDS (normal donkey serum, Abcam) in permeabilization solution. This was followed by addition of primary antibodies diluted in block. Details of the antibody dilutions and catalog information is given in Table 2. Primary antibody staining was carried overnight in 4 °C or 2 h in RT. After washing with 1XPBS, secondary antibody staining was carried for 45 min at RT. Nucleus was stained using 1XDAPI (Sigma). The slides were covered with Moviol (Sigma), mounted and sealed. All images were taken in FV3000 5 Laser confocal microscope.

## Quantification of IF images

To quantify the relative expression of the protein of interest in the control and the experimental samples, at least 3 biological replicates were immunostained strictly at the same time using identical protocol. Confocal microscopy images of the protein of interest were taken strictly at identical acquisition settings. The mean fluorescence intensity (MFI) of the epidermal compartment was ascertained using FIJI software in the entire epidermal area that was chosen manually. To quantify the expression of the protein of interest in the dermal macrophages, F4/80 immunostaining image was used to demarcate the boundary of macrophages. The expression of the protein of interest in the demarcated area was then ascertained using FIJI software. The MFI from each biological replicate was subsequently averaged to obtain an average MFI of the population. For each biological replicate, 50 macrophages were used for the quantification of MFI on an average. The extent of

basement membrane disruption was quantified manually by taking multiple measurements of the thickness of the basement membrane and averaging it for one biological replicate. Student unpaired t-test was performed on the averaged MFI from at least 3 biological replicate to quantify p values.

## Western blotting

Snap-frozen epidermis and dermis obtained from WT and cKO skin were pulverized using sterilized pestles. Homogenized tissue was then suspended in RIPA lysis buffer containing 1Xprotease inhibitor cocktail. Protein extraction was facilitated using multiple freeze–thaw cycles followed by centrifugation at maximum speed for 15 min at 4 °C. Protein concentration in the supernatant was measured using BCA assay (Promega). All protein isolate concentrations were normalized using RIPA-PIC buffer. 50 µg of protein was loaded onto PAGE (8%) and electrophoresed, and transferred onto PVDF membrane (BioRad). Blocking of the membrane was done using 5%BSA (Sigma). Primary antibody staining was done overnight at 4 °C. After washing with 0.1%TBST and secondary antibody (HRP conjugated) were added for an hour at RT. Unbound secondary antibodies were washed using 0.1% TBST and the blots were developed using ECL substrate (Thermo).

## RNA extraction and real-time PCR

Total RNA was extracted using Trizol (Thermo) for epidermal tissue and Trizol LS (Thermo) for sorted fibroblast and

**Table 2.** Table showing details of all antibodies used for the manuscript.

| S.No. | Antibody | Company | Catalog number | Dilution |
|---|---|---|---|---|
| 1. | Integrin Beta1 (ITGB1) | EMD Millipore | MAB1995 | 1:200 |
| 2. | Integrin Beta1 (ITGB4) | BD Biosciences | AB395027 | 1:200 |
| 3. | Glucose Transporter 1 (GLUT1) | Abcam | AB115730 | 1:300 |
| 4. | Lactate Dehydrogenase alpha (LDHa) | Abcam | AB52488 | 1:200 |
| 5. | Hexokinase 2 (HK2) | Cell Signalling Technology | C64G5 | 1:1000 |
| 6. | F4/80 | Ebioscience | 14-4801-82 | 1:200 |
| 7. | Isocitrate Dehydrogenase 1 (IDH1) | Abcam | AB172964 | 1:300 |
| 8. | Citrate Synthase (CS) | Novus Biologicals | NBP2-13878 | 1:300 |
| 9. | Laminin 332 (LAM332) | Gift by Bob Burgeson | – | 1:500 |
| 10. | Matrix Metalloproteinase 9 (MMP9) | R & D Systems | AF909 | 1:50 |
| 11. | Hypoxia Inducible Factor 1 alpha (HIF1α) | Novus Biologicals | NB100-449 | 1:200 |
| 12. | 8 hydroxy guanine (8-OHdG) | Novus Biologicals | NB100-1508 | 1:200 |
| 13. | E cadherin (ECAD) | Thermo Fisher Scientific | 131900 | 1:500 |
| 14. | Keratin 14 (KRT14) | Abcam | AB181595 | 1:1000 |
| 15. | Cyclooxygenase 2 (COX2) | Abcam | AB15191 | 1:200 |
| 16. | Monocarboxylic acid transporter 4 (MCT4) | Sigma | HPA021451 | 1:200 |
| 17. | Monocarboxylic acid transporter 1 (MCT1) | Sigma | PA5-72957 | 1:100 |
| 18. | PECAM (CD31) | Ebioscience | 14031181 | 1:200 |
| 19. | Alpha-Tubulin | Sigma | T9026 | 1:1000 |
| 20. | p65 NF-κB | CST | 93H1 | 1:500 |
| 21. | Glutathione | Abcam (Gifted by Mukherjee Lab, InStem) | 9443 | 1:100 |
| 22. | Catalase | Sigma (Gifted by Mukherjee Lab, InStem) | C0979 | 1:200 |

macrophages. Visualization of RNA pellet was facilitated by using 1 µL of Glycoblue (Thermo). Air-dried RNA was resuspended in nuclease free water (Thermo). Equal amounts of RNA obtained from cKO and control skin compartments were used to prepare cDNA by using SSIII RT cDNA synthesis kit (Invitrogen). SYBR green (2X) master mix (Invitrogen) was used for real-time PCR. Delta Ct method was used to quantify the relative changes in the level of transcripts.

18S was used as an endogenous control. The list of primers used is given in Table 3.

## Metabolomics

### Material

TCA cycle (ML0010) and glycolysis/gluconeogenesis (ML0013) metabolite libraries were obtained from Sigma-Aldrich (St. Louis, MO). Isotopic 13C15N-labeled amino acids (MSK-CAA-1) were obtained from Cambridge Isotope Laboratories (Andover, MA). Optima-LC/MS grade isopropanol (A461-4) and acetonitrile (A955-4) were obtained from ThermoFisher Scientific (Waltham, MA).

### Methodology

**Tissue sample processing** A total of 44 pre-weighed samples (22 epidermis, 22 dermis) were randomized and thawed prior to processing. To each sample, a volume of 300 µL of Type 1 ultrapure water was added and bead-homogenized at 4500 rpm, $3 \times 10$ s cycles at 4 °C on a Cryolys Evolution-Precellys Evolution tissue homogenizer (Rockville, MD). A volume of 100 µL of tissue homogenate was added to 500 µL ice-cold isopropanol containing 1.25 µM of 13C15N-labeled amino acids internal standards, and agitated for 1 h to promote protein precipitation. Samples were then centrifuged at $20,000 \times g$ and the supernatant was collected. Extraction was repeated with another portion of 500 µL ice-cold isopropanol with internal standards. Pooled supernatants were then evaporated on a Labconco CentriVap concentrator (Kansas, MO). Samples were reconstituted in 100 µL acetonitrile:water (1:1), centrifuged at $14,000 \times g$ at 4 °C for 10 min, then introduced into the liquid chromatography-tandem mass spectrometry (LC-MS/MS) at injection volume of 1 µL.

**Liquid chromatography mass spectrometry** Multiple reaction monitoring was performed on a Thermo Scientific Vanquish Duo UHPLC coupled to a Quantis TSQ Triple Quadrupole Mass Spectrometer. Chromatographic separation was done on a Waters Atlantis Premier BEH Z-HILIC column (1.7 µm, 2.1 mm × 100 mm) maintained at 45 °C. Chromatographic separation was achieved at 0.45 mL/min using the gradient elution as follows: 0.0 min 99%B, 15.0 min 30%B, 17.9 min 30%B, 18.0 min 99%B, 25.0 min 99%B. Mass spectrometry was operated in electrospray ionization (ESI) mode with polarity switching, with source parameters as follows: spray voltage 3.5 kV (positive) −2.8 kV (negative), sheath gas 50 [arbitrary units], auxiliary gas 18 [arbitrary units], sweep gas 0 [arbitrary units], ion transfer tube temperature 325 °C, vaporizer temperature 230 °C. Details on mobile phases and compound-specific parameters are described in the Supplementary.

**Table 3.** List of all primer sequenced used for the manuscript.

| Gene name | Forward primer | Reverse primer |
| --- | --- | --- |
| Serpine1 | TTCAGCCCTTGCTTGCCTC | ACACTTTTACTCCGAAGTCGGT |
| Vegfa | GCACATAGAGAGAATGAGCTTCC | CTCCGCTCTGAACAAGGCT |
| Timp1 | GCAACTCGGACCTGGTCATAA | CGGCCCGTGATGAGAAACT |
| Fn1 | ATGTGGACCCCTCCTGATAGT | GCCCAGTGATTTCAGCAAAGG |
| Id2 | ATGAAAGCCTTCAGTCCGGTG | AGCAGACTCATCGGGTCGT |
| Krt14 | AGCGGCAAGAGTGAGATTTCT | CCTCCAGGTTATTCTCCAGGG |
| IGFBP3 | CCAGGAAACATCAGTGAGTCC | GGATGGAACTTGGAATCGGTCA |
| Gpx3 | TTTGTGCCTAATTTCCAGCTCTT | GTCCATCTTGACGTTGCTGAC |
| H6PD | ATGAAGCACACAGGCATTTGG | TCCAGGTATAGCTGAAACAGTCC |
| Nqo1 | AGGATGGGAGGTACTCGAATC | AGGCGTCCTTCCTTATATGCTA |
| Prdx6 | GTCGAGAAGGACGCTAACAAC | GGGTAGAGGATAGACAGCTTCAG |
| Alox5 | TTGCTCTCACAGTATGACTGGT | AGTATCCACGATCTGCTCGA |
| Alox8 | CTGTCAGCATCGTGGGAACC | GGAAGCGTCACCTCGAAGTC |
| Duox1 | AAAACACCAGGAACGGATTGT | AGAAGACATTGGGCTGTAGGG |
| DuoxA1 | ACCAAGCCAACCTTTCCAATG | GCCCCGATGAATAAGCTGGTC |
| Mpo | AGTTGTGCTGAGCTGTATGGA | CGGCTGCTTGAAGTAAAACAGG |
| Xdh | ATGACGAGGACAACGGTAGAT | TCATACTTGGAGATCATCACGGT |
| GLUT1 | CAGTTCGGCTATAACACTGGTG | GCCCCCGACAGAGAAGATG |
| Hk2 | TGATCGCCTGCTTATTCACGG | AACCGCCTAGAAATCTCCAGA |
| LDHa | CATTGTCAAGTACAGTCCACACT | TTCCAATTACTCGGTTTTTGGGA |
| PKM2 | GCCGCCTGGACATTGACTC | CCATGAGAGAAATTCAGCCGAG |
| MCT4 | TCACGGGTTTCTCCTACGC | GCCAAAGCGGTTCACACAC |
| Pfkfb3 | CCCAGAGCCGGGTACAGAA | GGGGAGTTGGTCAGCTTCG |
| ME2 | AAGGGAATGGCGTTTACGTTAC | GTACACAATCGGCATCAGACTT |
| Tpi | CCAGGAAGTTCTTCGTTGGGG | CAAAGTCGATGTAAGCGGTGG |
| Enolase 1 | TGCGTCCACTGGCATCTAC | CAGAGCAGGCGCAATAGTTTTA |
| Ogdh | GGAACTGCCCTCTAGGGAGA | GACGCTACCACTGTTAATGACC |
| FH1 | GAATGGCAAGCCAAAATTCCTT | CGTTCTGTAGCACCTCCAATCTT |
| Cs | GGACAATTTTCCAACCAATCTGC | TCGGTTCATTCCCTCTGCATA |
| Aco1 | AGAACCCATTTGCACACCTTG | AGCGTCCGTATCTTGAGTCCT |
| Aco2 | ATCGAGCGGGGAAAGACATAC | TGATGGTACAGCCACCTTAGG |
| Idh1 | ATGCAAGGAGATGAAATGACACG | GCATCACGATTCTCTATGCCTAA |
| Idh2 | CACCGTCCATCTCCACTACC | CAGCACTGACTGTCCCCAG |
| Got2 | TGGGCGAGAACAATGAAGTGT | CCCAGGATGGTTTGGGCAG |
| Sdha | GGAACACTCCAAAAACAGACCT | CCACCACTGGGTATTGAGTAGAA |
| Sdhb | GCTGCGTTCTTGCTGAGACA | ATCTCCTCCTTAGCTGTGGTT |
| Sdhc | TGGTCAGACCCGCTTATGTG | GGTCCAGTGGAGAGATGCAG |
| Sdhd | CGAAAGCGACATGGCGGTTC | GGTCCTGGAGAAATGCTGACAC |
| Pfkl | GGAGGCGAGAACATCAAGCC | CGGCCTTCCCTCGTAGTGA |

## Statistical analysis

All the statistical analysis presented in the paper were done in GraphPad Prism version 9.0.0. Two-tailed students t-test has been performed across all graphs with at least 3 biological replicates.

## Data availability

RNA sequencing used in the report has been done previously (Bhattacharjee et al, 2021). The data sets obtained from the report are submitted in NCBI with reference ID: SRP324814

(PRJNA739149). The data can additionally be accessed from the link: https://dataview.ncbi.nlm.nih.gov/object/PRJNA739149?reviewer=1klobv7agr78somt2toldvkjvu.

## Peer review information

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

## Acknowledgements

We would like to thank Dr. Sunil Laxman and Dr. Tina Mukherjee (InStem, Bangalore) for sharing lab reagents and providing critical feedback for the work conducted. We would additionally like to thank Dr. Dasaradhi Palakodeti (InStem, Bangalore) and Dr. Ramanuj Dasgupta (Genome Institute of Singapore), and members of the Raghavan and Mukherjee Lab for providing critical inputs for our work and giving valuable feedback on the work and manuscript. We thank Dr. Roopa from ACRC facility at NCBS for training us in mouse in utero surgeries. This work is supported by the grant from the Department of Biotechnology, (DBT), India, DBT grant BT/PR31418/BRB/10/1758/2019, Institute for Stem Cell Biology and Regenerative Medicine (InStem), India core funding to SR and by the AMBM grant (A18A8b0059) (supporting SR in Singapore). UA is funded through a DBT predoctoral fellowship DBT/JRF/BET-18/1/2018/AL/60. Animal work was partially supported by the National Mouse Research Resource (NaMoR) grant (BT/PR5981/MED/31/181/2012; 2013-2016) from the DBT. JC, SZE, AFBA, HT and AK are supported through core and grant funding from the Agency for Science, Technology and Research (A*STAR) under its Industry Alignment Fund – Pre-Positioning Programme (IAF-PP) as part of the Asian Skin Microbiome Programme 2.0 (ASMP 2.0, grant number H22J1a0040).

## Author contributions

**Uttkarsh Ayyangar**: Conceptualization; Data curation; Formal analysis; Validation; Investigation; Visualization; Methodology; Writing—original draft; Writing—review and editing. **Aneesh Karkhanis**: Formal analysis; Validation; Investigation; Methodology. **Heather Tay**: Formal analysis; Validation; Investigation; Methodology. **Aliya Farissa Binte Afandi**: Formal analysis; Validation; Investigation; Methodology. **Oindrila Bhattacharjee**: Methodology. **Lalitha KS**: Methodology. **Sze Han Lee**: Resources; Data curation; Formal analysis; Validation; Investigation; Methodology; Writing—original draft. **James Chan**: Resources; Formal analysis; Supervision; Validation; Investigation;

Methodology; Writing—original draft. **Srikala Raghavan**: Conceptualization; Resources; Data curation; Formal analysis; Supervision; Funding acquisition; Validation; Investigation; Methodology; Writing—original draft; Project administration; Writing—review and editing.

## Disclosure and competing interests statement
The authors declare no competing interests.

