## [Peer Review File · The EMBO Journal]

Metabolic rewiring of macrophages by epidermal-derived lactate promotes sterile inflammation in the murine skin

Uttkarsh Ayyangar, Aneesh Karkhanis, Heather Tay, Aliya Afandi, Oindrila Bhattacharjee, Lalitha KS, Sze Han Lee, James Chan, and Srikala Raghavan

DOI: [10.15252/emboj.2023113774](https://doi.org/10.15252/emboj.2023113774)

Corresponding authors: Srikala Raghavan (srikala_raghavan@asrl.a-star.edu.sg) , Uttkarsh Ayyangar (uttkarsha@instem.res.in)

Review Timeline:

Submission Date:	15th Feb 23
Editorial Decision:	28th Apr 23
Appeal:	26th Nov 23
Editorial Decision:	20th Dec 23
Revision Received:	2nd Jan 24
Accepted:	8th Jan 24

Editor: Daniel Klimmeck

Transaction Report:

Dear Dr Raghavan,

Thank you for resubmission of your manuscript EMBOJ-2023-113774. Please accept my sincere apologies for getting back to you with this unusual protraction. We have sent your manuscript to assessment by three referees with expertise in skin biology, inflammation and metabolism, however one referee got substantially delayed and, in the end, did not provide his/her report even after repeated chasers. We have now decided to base our decision on the two existing reports which are enclosed below. I regret to say that the outcome of our considerations is that we cannot offer to proceed with peer-review.

As you will see, both reviewers consistently acknowledge the interest in your findings on metabolic compartmentalisation of the epidermis and a respective interplay between skin cells and macrophages. However, they also point to substantial concerns on the functional support for the claims made and overall robustness of the findings (see ref#1, pts.1; 2-4) as well as mechanistic depth of insights and remaining ambiguities (ref #2, pts.1,2).

We have discussed all information at hand carefully in the editorial team and considering the current data presented concluded that the key claims are insufficiently supported by the data, and that the outcome of complementary experimentation is entirely open at this stage. We have thus decided to formally reject the manuscript at this stage.

Please note however, that we would be able to assess a substantially amended version of the manuscript if the critique could be conclusively addressed and the proposed model holds up.

I regret that I do not have more positive information to be shared. I again apologise for the protraction.

Kind regards,

Daniel Klimmeck

Daniel Klimmeck, PhD
Senior Editor
The EMBO Journal

Referee #2:

The manuscript by Ayyangar et al. "Epidermal directed metabolic rewiring of macrophages promotes sterile inflammation in murine skin" describes experiments undertaken to dissect the metabolic regulation of macrophage-driven skin inflammation and determine how epidermal cell metabolic activity shapes the local milieu of metabolic substrates available for macrophages. The authors propose that metabolic changes in the epidermis dictate the inflammatory phenotype of macrophages based on substrate availability. The authors observe that during skin inflammation, epidermal cells rely on glycolysis while macrophages rely on the TCA cycle. The authors observe HIF1a stabilization specifically in the epidermis which in turn promotes glycolysis and propose that this provides lactate to macrophages, supporting their inflammatory effector functions. They then expand their studies to the imiquimod model of psoriatic inflammation where they further test the role of lactate in skin inflammation. This question of compartmentalization of metabolic functions between distinct subsets of cells in an inflamed tissue and how it regulates pathology is very interesting, however many of the precise claims made by the authors lack experimental support.

Major Criticisms

1) The experiments performed do not prove a direct crosstalk loop between epidermal cells and macrophages. Immunofluorescence analysis is primarily used to show that specific transporters or metabolic enzymes are expressed in either macrophages or epidermal cells, and then whole-body inhibition of specific pathways is used to inhibit those pathways. While this does argue that those pathways are functional and are important for tissue inflammation, it does not exclude the possibility that metabolic substrates from circulation are impacting macrophage function and thus strictly speaking it cannot be claimed that there is local, direct, epidermal/macrophage crosstalk.

2) The methods underlying immunofluorescence microscopy analysis are not described in sufficient detail to be confident in the results presented. Given how critical IF data is to the claims made in this manuscript, it is crucial that the manuscript describe precisely how images were acquired (were consistent acquisition settings used? How were regions of the sections identified for mean fluorescence intensity quantification? Was MFI calculated from individual positive cells or was it determined as the average across the entire section?). In particular, Figure 5J,K claims to quantify MFI of IDH1 or CS within macrophages but

there is no mention in the methods of how this "within macrophages" quantification was performed. Was any algorithm or program used to process the data?

3) Many claims in the manuscript rely on only a single type of analysis. The vast majority of critical data in this manuscript are derived from IF imaging with minimal experimental validation using alternative methods (cell sorting followed by qPCR, etc.). For example, dispase digestion can be used to separate dermal and epidermal sheets from the mouse ear which could then be subjected to qPCR, western blot, or flow cytometry analysis to strengthen the claims about compartmentalized metabolic functions.

4) The sample size for many experiments is insufficient to support the claims made and, in many cases, improper statistical analyses have been applied. A student's t-test is used for all statistical analysis presented in the manuscript. Data evaluated by t-test must be drawn from populations that are normally distributed and with roughly equal variances, when sample size is <3 (in many cases $N=2$ is presented in this paper) it cannot be assumed that the data meets the requirements for a t-test to be the appropriate statistical test. Additionally, performing multiple pairwise t-tests on a single complex data set (two conditions, multiple timepoints, etc) massively inflates the risk of falsely deeming a non-significant finding as significant. When multiple hypotheses are tested from a single data set one should either use an ANOVA (one-way or two-way depending on the nature of the dataset) or apply some sort of multiple hypothesis testing correction to the significance threshold from the t-test (such as a Bonferroni correction). Furthermore, it is unclear whether technical or biological replicates were used to perform the statistical analysis shown in several places. In panels 1I; 2A; 3G; 4 C,H-J, 5D,G,H,J,K; 6B,C,H; 7D, and 8H,I the statistical significance indicated by stars above the plot seems unrealistic given the effect size and error bars on the plot and the number of biological replicates indicated in the figure legend. 7D is a particularly clear example of this, the difference in the means is probably about 20%, with error bars spanning a range of roughly $\pm 50\%$ of the group means with $N=3$ and the p-value is indicated to be less than 0.01. This is simply not realistic given the indicated number of replicates and the data spread indicated by the plot. Were many technical replicates from the 2 or 3 biological samples used to run these analyses? If so, it would be more appropriate to average the technical replicates and run the t-test on the actual biological mean values. For 4H, the legend does indicate the 30-40 cells from $N=2$ were analyzed, were all 30-40 cells per group used to generate the plot shown and perform statistical analysis or were the cells from each mouse sample averaged and then those data plotted? The former would inflate the statistical significance of the observed biological effect. Changing the data presentation to one where individual data points are shown on top of the bar plot would make it much easier to understand the true difference between the groups.

Referee #3:

This study reports that the epidermis and macrophages adopt different metabolic mechanisms during sterile inflammation in the skin - glycolysis and TCA, respectively. Lactate, as the end product of glycolysis from the epidermis, influences macrophage metabolism and functions in the dermis. The authors also provide evidence to link psoriatic with lactate-mediated crosstalk between the epidermis and macrophages. In general, this is an interesting finding with smartly designed experiments and valuable speculations for future research. Here are some questions the authors should consider in revising the manuscript.

1. The authors cited previous research that in vitro polarised M1 macrophages adopt glycolysis and M2 cells utilise TCA as the main mode of metabolism. But, here the authors found that during sterile inflammation macrophages in the dermis rely on TCA. Meanwhile, we know that the in vitro M1/M2 nomenclature is oversimplified. So, some additional data are needed for several questions around the phenotype of macrophages in the model used in this study:

a) What are the phenotypes of the dermal macrophages in this model? More M2-like? How heterogeneous are they?
b) Is the current finding contradictory with in vitro findings - or does it partly reflect a stage of a dynamic change in the phenotype of dermal macrophages under sterile inflammation? If so, some tests at different time points?
c) Is it possible that different types, subsets or locations of macrophages uptake lactate differently? Is there any direct imaging evidence to localise lactate distribution and differently polarised macrophages with specific markers? Do tissue-resident macrophages and infiltrated monocytes (and perhaps other myeloid cells) respond similarly to lactate? Do macrophages secrete metabolites to 'counter-affect' the epidermis? A more in-depth depiction of this process is required.

2. Even if increased lactate is produced and exported to the dermis, what mechanism prompts macrophages to decide to 'use' lactate as the major carbon source if they were not doing it originally? The simplest answer could be because of the abundance of lactate in the niche; however, alongside lactate there must be other pro-inflammatory molecules released from the epidermis. Since macrophages are highly plastic, there must be a regulatory mechanism balancing phenotype maintenance and metabolic change. Is there a model that macrophages lack the ability to uptake or metabolize lactate (e.g. MCT1 ko?) with which we could obtain more information?

3. The finding that the loss of epidermal integrin b1 led to ROS upregulation is interesting but could have been studied more clearly. Itgb1 is known as a stemness marker for basal keratinocytes (epidermal stem cells). It also plays a key role in mediating cell-matrix interaction which is crucial for the survival of many anchorage-dependent cells. So, more data could have been provided on how did its loss lead to ROS increase, e.g. feedback to resist apoptosis? A consequence of differentiation/stratification? Or a consequence of tightening of cell-cell adhesion?

4. Which exact group of keratinocytes are producing lactate under sterile inflammation? How about the duration and kinetics of their lactate production? What is the fate of these lactate-producing keratinocytes? (e.g. increase in involucrin, apoptosis or

stratified?)

5. What are the main metabolic mechanism for dermal macrophages without induced inflammation? What metabolic change will happen if lactate is directly injected to the dermis in either embryos or adult mice?

6. Minor suggestions:

a) Metabolic pathway-related gene expression was tested and clustered. But for some key intermediate products, biochemical assays of quantification could be added for validation

b) In several figures red shadows (or bushes) are used to depict ECM degradation, which can be confusing. Perhaps another way of illustration?

c) The effectiveness of treating pregnant dams with agents on the embryo could be discussed

d) Provide more discussion on the implication of this finding for other autoimmune diseases.

** As a service to authors, EMBO Press provides authors with the possibility to transfer a manuscript that one journal cannot offer to publish to another EMBO publication or the open access journal Life Science Alliance launched in partnership between EMBO Press, Rockefeller University Press and Cold Spring Harbor Laboratory Press. The full manuscript and if applicable, reviewers' reports, are automatically sent to the receiving journal to allow for fast handling and a prompt decision on your manuscript. For more details of this service, and to transfer your manuscript please click on Link Not Available. **

Referee #2:

The manuscript by Ayyangar et al. "Epidermal directed metabolic rewiring of macrophages promotes sterile inflammation in murine skin" describes experiments undertaken to dissect the metabolic regulation of macrophage-driven skin inflammation and determine how epidermal cell metabolic activity shapes the local milieu of metabolic substrates available for macrophages. The authors propose that metabolic changes in the epidermis dictate the inflammatory phenotype of macrophages based on substrate availability. The authors observe that during skin inflammation, epidermal cells rely on glycolysis while macrophages rely on the TCA cycle. The authors observe HIF1a stabilization specifically in the epidermis which in turn promotes glycolysis and propose that this provides lactate to macrophages, supporting their inflammatory effector functions. They then expand their studies to the imiquimod model of psoriatic inflammation where they further test the role of lactate in skin inflammation. This question of compartmentalization of metabolic functions between distinct subsets of cells in an inflamed tissue and how it regulates pathology is very interesting, however many of the precise claims made by the authors lack experimental support.

We thank the reviewer for their comments. In this substantially revised manuscript, we have added additional data to support our claims and edited the graphs as suggested. We have additionally increased the number of biological replicates for each experiment to at least 3 to strengthen our conclusions. The revisions suggested by the reviewer have substantially improved the manuscript.

Major Criticisms

1) The experiments performed do not prove a direct crosstalk loop between epidermal cells and macrophages. Immunofluorescence analysis is primarily used to show that specific transporters or metabolic enzymes are expressed in either macrophages or epidermal cells, and then whole-body inhibition of specific pathways is used to inhibit those pathways. While this does argue that those pathways are functional and are important for tissue inflammation, it does not exclude the possibility that metabolic substrates from circulation are impacting macrophage function and thus strictly speaking it cannot be claimed that there is local, direct, epidermal/macrophage crosstalk.

We agree with the reviewer that intraperitoneal injection of the drugs to inhibit lactate transporters do not necessarily suggest a direct crosstalk between epidermis and macrophages. Hence, we have performed additional experiment to test the idea of a local crosstalk between the two compartments in the skin. To show this in the embryonic skin, we have administered the lactate transport inhibitor, Syrosingopine, *in utero* into the amniotic sac at embryonic day E15.5 to target the drug specifically into the epidermal compartment. This strategy has been used by us previously in several studies to target drugs and antibodies into the skin to study key regulators of skin development (Kurbet *et al*, 2016). Our results show that targeting Syrosingopine to the epidermal compartment leads to a remarkable attenuation of the M2-like pro-ECM remodeling phenotype of dermal macrophages in the cKO skin. This is characterized by a reduction in basement membrane degradation and expression of ECM remodeling enzyme MMP9 compared to the controls. While this data does not negate the possibility of systemic metabolites having a role in driving macrophage function, it provides evidence that direct lactate-mediated crosstalk between the epidermis and macrophages is sufficient to drive macrophage effector states in chronic skin inflammation. The data for this has been provided in Figure 5 (G and H).

The schematic representation of the intra-uterine dosage has been provided in Supplementary Figure S7.

Furthermore, to show that lactate alone is sufficient to MMP9 synthesis by the macrophages, we injected sodium-lactate to the back skin of neonatal mice at neonatal

stage P0 though the subcutaneous route. The skin was subsequently extracted at post-natal day P2. We observed a significant increase in the expression of MMP9 in the back skin of the mice treated with sodium-lactate compared to controls. This suggests that lactate alone is sufficient to increase the expression of MMP9 in the skin. The data for this experiment is provided in Figure 5 (A and B).

2) The methods underlying immunofluorescence microscopy analysis are not described in sufficient detail to be confident in the results presented. Given how critical IF data is to the claims made in this manuscript, it is crucial that the manuscript describe precisely how images were acquired (were consistent acquisition settings used? How were regions of the sections identified for mean fluorescence intensity quantification? Was MFI calculated from individual positive cells or was it determined as the average across the entire section?). In particular, Figure 5J,K claims to quantify MFI of IDH1 or CS within macrophages but there is no mention in the methods of how this "within macrophages" quantification was performed. Was any algorithm or program used to process the data?

We have added a detailed methods section (Please see Methods: Quantification of IHC images) to answer the questions raised by the reviewer. To quantify the relative expression of the protein of interest in the control and the experimental samples, at least 3 biological replicates were immunostained strictly at the same time using identical protocol. Confocal microscopy images of the protein of interest were taken strictly at identical acquisition settings. The mean fluorescence intensity (MFI) of the epidermal compartment was ascertained using FIJI software in the entire epidermal area that was chosen manually. To quantify the expression of the protein of interest in the dermal macrophages, F4/80 immunostaining image was used to demarcate the boundary of macrophages. The expression of the protein of interest in the demarcated area was then ascertained using FIJI software. The MFI from each biological replicate was subsequently averaged to obtain an average MFI of the population. For each biological replicate, 50 macrophages were used for the quantification of MFI on an average. The extent of basement membrane disruption was quantified manually by taking multiple measurements of the thickness of the basement membrane and averaging it for one biological replicate.

Student unpaired t-test was performed on the averaged MFI from at least 3 biological replicate to quantify p values. These changes have been incorporated for all the graphs across all figures.

3) Many claims in the manuscript rely on only a single type of analysis. The vast majority of critical data in this manuscript are derived from IF imaging with minimal experimental validation using alternative methods (cell sorting followed by qPCR, etc.). For example, dispase digestion can be used to separate dermal and epidermal sheets from the mouse ear which could then be subjected to qPCR, western blot, or flow cytometry analysis to strengthen the claims about compartmentalized metabolic functions.

IF imaging has been primarily used to show the extent of basement membrane disruption, generation of MMP9, and localization of metabolic enzymes in different compartments of the skin. As suggested by the reviewer, we have made the following additions to the manuscript to strengthen our claims:

1. We have performed qPCRs and western blot for quantifying the expression of glucose transporters and glycolytic enzymes in the epidermal compartment of the skin which was isolated after dispase digestion (Figure 1A-D). In addition to this, we have performed steady state metabolomics to quantify the relative change in the levels of glycolysis and TCA metabolites in the epidermal compartment of the skin (Figure 1E and F). These data sets support the initial hypothesis of a glycolytic switch in the epidermal compartment which was drawn out after looking at the NGS data (Supplementary Figure 1).

2. We have done qPCR for HIF1 α target genes in the isolated epidermal compartment from the cKO and WT skin to show increase in the transcription of downstream

targets of HIF1 (Figure 2D). We further show using qPCR that HIF1 inhibition using chetomin reduces the transcription of glycolysis targets in the epidermis of the cKO skin compared to controls (Figure 2E). We have additionally added the list of HIF1 associated genes and their fold changes in the NGS data obtained from the isolated epidermal compartment of the cKO skin compared to controls (Supplementary Figure S2 A).

3. We have added the qPCR for ROS source and scavenger genes in the epidermal compartment of the skin to show increased transcription in the cKO skin (Figure 3 D and E). We have further added immunostaining for Glutathione and catalase to show increased expression of anti-oxidants in the cKO skin (Figure 3F).

4. For the macrophage compartment, we have performed qPCR for glycolysis and TCA cycle associated genes in FACS sorted macrophages obtained from the cKO and WT

skin (Supplementary Figure S4a D and F). In addition, we have performed steady state metabolomics for the dermal compartment of cKO skin (Figure 4 D and E). These results corroborate the NGS data obtained from FACS sorted macrophages (Supplementary Figure S4a).

- We have performed qPCR of psoriatic biomarkers in the imiquimod skin treated with lactate inhibitor Syrosingopine and DMSO controls to show decreased transcription of psoriatic biomarkers after lactate transport inhibition (Figure 8I).

Taken together, in the revised manuscript we have used qPCR, western blot, and steady state metabolomics in addition to IHC quantification to support our claims. These changes have significantly improved the rigor of the manuscript.

4) The sample size for many experiments is insufficient to support the claims made and, in many cases, improper statistical analyses have been applied. A student's t-test is used for all statistical analysis presented in the manuscript. Data evaluated by t-test must be drawn from populations that are normally distributed and with roughly equal variances, when sample size is <3 (in many cases N=2 is presented in this paper) it cannot be assumed that the data meets the requirements for a t-test to be the appropriate statistical test. Additionally, performing multiple pairwise t-tests on a single complex data set (two conditions, multiple timepoints, etc) massively inflates the risk of falsely deeming a non-significant finding as significant. When multiple hypotheses are tested from a single data set one should either use an ANOVA (one-way or two-way depending on the nature of the dataset) or apply some sort of multiple hypothesis testing correction to the significance threshold from the t-test (such as a Bonferroni correction). Furthermore, it is unclear whether technical or biological replicates were used to perform the statistical analysis shown in several places. In panels 1I; 2A; 3G; 4 C,H-J, 5D,G,H,J,K; 6B,C,H; 7D, and 8H,I

the statistical significance indicated by stars above the plot seems unrealistic given the effect size and error bars on the plot and the number of biological replicates indicated in the figure legend. 7D is a particularly clear example of this, the difference in the means is probably about 20%, with error bars spanning a range of roughly +/- 50% of the group means with N=3 and the p-value is indicated to be less than 0.01. This is simply not realistic given the indicated number of replicates and the data spread indicated by the plot. Were many technical replicates from the 2 or 3 biological samples used to run these analyses? If so, it would be more appropriate to average the technical replicates and run the t-test on the actual biological mean values. For 4H, the legend does indicate the 30-40 cells from N=2 were analyzed, were all 30-40 cells per group used to generate the plot shown and perform statistical analysis or were the cells from each mouse sample averaged and then those data plotted? The former would inflate the statistical significance of the observed biological effect. Changing the data presentation to one where individual data points are shown on top of the bar plot would make it much easier to understand the true difference between the groups.

In the revised manuscript, we have taken the average of intensities obtained from multiple images obtained from each biological replicate to plot the graphs. For all data sets, we now have at least 3 biological replicates to support our claims. We have used only biological replicates, and not technical replicates, of at least 3 to calculate statistical significance. All the graphs have thus been modified. To the best of our knowledge, we have applied appropriate statistical tests to ascertain the statistical significance of the data.

Referee #3:

This study reports that the epidermis and macrophages adopt different metabolic mechanisms during sterile inflammation in the skin - glycolysis and TCA, respectively. Lactate, as the end product of glycolysis from the epidermis, influences macrophage metabolism and functions in the dermis. The authors also provide evidence to link psoriatic with lactate-mediated crosstalk between the epidermis and macrophages. In general, this is an interesting finding with smartly designed experiments and valuable speculations for future research. Here are some questions the authors should consider in revising the manuscript.

We thank the reviewer for their constructive comments. In the revised manuscript, we have incorporated most of the suggestions given by the reviewer. We believe that these revisions have significantly improved the rigor of the manuscript.

1. The authors cited previous research that in vitro polarised M1 macrophages adopt glycolysis and M2 cells utilise TCA as the main mode of metabolism. But, here the authors found that during sterile inflammation macrophages in the dermis rely on TCA. Meanwhile, we know that the in vitro M1/M2 nomenclature is oversimplified. So, some additional data are needed for several questions around the phenotype of macrophages in the model used in this study:

a) What are the phenotypes of the dermal macrophages in this model? More M2-like? How heterogeneous are they?

We agree with the reviewer that M1/M2 nomenclature is rather simplified and are derived from in vitro experiments. That said, the pro-remodelling macrophages exists in a

continuum of M2 states and to reflect that we refer to these as "M2-like" throughout the manuscript. We have previously addressed the question of macrophage functional state and heterogeneity in the a paper published from our lab (Bhattacharjee *et al*, 2021). In the paper we show that macrophages in the cKO skin are primarily maintained in an M2-like state which is characterized by enhanced expression of classical M2 markers such as CD206 and MERTK and increased generation of ECM remodelling enzyme MMP9 (Bhattacharjee *et al*, 2021). We have shown that in addition to these M2-like macrophages, there is a small percentage monocyte (CD11B+ve cells) and a very small population of M1-like macrophages (CD38+ve) as well. These data address the concern of macrophage heterogeneity in the cKO skin. Given these data, we have focussed our attention to the M2-like macrophages in the cKO skin for this manuscript.

b) Is the current finding contradictory with in vitro findings - or does it partly reflect a stage of a dynamic change in the phenotype of dermal macrophages under sterile inflammation? If so, some tests at different time points?

The macrophages in the cKO skin increase dependence on TCA cycle metabolism to drive their M2-like effector state. This is in line with the in-vitro studies that suggest that IL-4/13 treated M2 macrophages have an intact TCA metabolism. Our data, in fact, do not contradict the existing in vitro findings.

c) Is it possible that different types, subsets or locations of macrophages uptake lactate differently? Is there any direct imaging evidence to localise lactate distribution and differently polarised macrophages with specific markers? Do tissue-resident macrophages and infiltrated monocytes (and perhaps other myeloid cells) respond similarly to lactate? Do macrophages secrete metabolites to 'counter-affect' the epidermis? A more in-depth depiction of this process is required.

We think that this is an interesting idea. Evidence in the existing literature suggest that circulating monocytes are highly dependent on glycolysis (Lee *et al*, 2019). Whereas, tissue resident macrophages, particularly in the M2 state, prefer the TCA cycle (Kelly & O'Neill, 2015). While we would have liked to address the question of localized lactate distribution in the in the macrophage subpopulations, the direct imaging of lactate is not possible due to resolution limitations of the available spatial metabolomics platforms (~25 microns). In this revised manuscript we have shown that exogenous lactate induces MMP9 synthesis in neonatal mice back skin (Figure 5A and B). The fact that exogenous administration of lactate into the skin was sufficient to repolarize the tissue resident macrophages to an M2-like pro-ECM remodelling state suggests that infiltrating monocytes may also respond to increased local lactate levels potentially by differentiating into M2-like macrophages.

Our unpublished data, not shown in the paper, suggests that there is active feedback from the macrophages to the epidermis. Our results show that if we prevent macrophages from importing lactate, there is a resultant reduction in the expression of glucose transporters in the epidermis and increases TCA cycle expression. We are currently pursuing this avenue. The data for this is shown below.

2. Even if increased lactate is produced and exported to the dermis, what mechanism prompts macrophages to decide to 'use' lactate as the major carbon source if they were not doing it originally? The simplest answer could be because of the abundance of lactate in the niche; however, alongside lactate there must be other pro-inflammatory molecules released from the epidermis. Since macrophages are highly plastic, there must be a regulatory mechanism balancing phenotype maintenance and metabolic change. Is there

a model that macrophages lack the ability to uptake or metabolize lactate (e.g. MCT1 ko?) with which we could obtain more information?

We agree that in addition to niche metabolites, there are cytokines generated by the epidermis which drives macrophages effector function in the cKO skin. We have previously shown that blocking these cytokines are sufficient to attenuate macrophage function in the cKO skin (Bhattacharjee *et al*, 2021; Kurbet *et al*, 2016). In the current manuscript, we show that epidermally-derived lactate further drives macrophage effector state in the cKO skin. We agree with the reviewer that increase in the local levels of lactate potentially prompts the macrophages to utilize it. To test this idea, we injected sodium-lactate into the back skin of neonatal mice at P0 stage through the subcutaneous route. The skin was subsequently extracted at post-natal day P2. We observed a significant increase in the expression of MMP9 in the back skin of the mice treated with sodium-lactate compared to controls. This suggests that lactate alone is sufficient to increase the expression of MMP9 in the skin. The data for this experiment is provided in Figure 5 (A and B). Furthermore, we have blocked the release of lactate by the epidermal compartment of the skin by injecting Syrosingopine *in-utero* into the amniotic sac (Supplementary figure S7). The results show that *in-utero* Syrosingopine administration prevents the polarization of the dermal macrophages into an M2-like state in cKO skin presumably due to the inability of macrophages to uptake lactate from the niche (Figure 5D and E).

3. The finding that the loss of epidermal integrin b1 led to ROS upregulation is interesting but could have been studied more clearly. Itgb1 is known as a stemness marker for basal keratinocytes (epidermal stem cells). It also plays a key role in mediating cell-matrix interaction which is crucial for the survival of many anchorage-dependent cells. So, more data could have been provided on how did its loss lead to ROS increase, e.g. feedback to resist apoptosis? A consequence of differentiation/stratification? Or a consequence of tightening of cell-cell adhesion?

We agree with the reviewer that this is an important question to address in the manuscript. Loss of integrin beta1 leads to the detachment of the epidermis from the underlying basement membrane. This is potentially observed as a mechanical stress in the skin which has been shown to increase ROS. To further strengthen the idea of mechanical stress induced ROS upregulation, we further show that loss of integrin beta1 leads to aberrant expression of integrin b4 and increased the expression of epidermal stress keratin, keratin 6. This additionally increases the expression of mechanical stress induced ECM component, Tenascin C. Importantly, the increase in these stress induced markers happen early on at embryonic day E16.5. The data has been added in Supplementary Figure S3A.

4. Which exact group of keratinocytes are producing lactate under sterile inflammation? How about the duration and kinetics of their lactate production? What is the fate of these lactate-producing keratinocytes? (e.g. increase in involucrin, apoptosis or stratified?)

To answer this, we have co-stained the cKO skin with the basal cell marker KRT5 and MCT4. The results show that the basal keratinocytes primarily have increased membrane expression of MCT4 transporter suggesting that basal keratinocytes are the primary exporters of lactate in the cKO skin (Supplementary Figure S4a A). By temporal analysis we further show that the membrane expression of MCT4 augments around embryonic day E17.5 and continues till E18.5 stage (Figure 4A). We have shown previously that the loss of integrin beta1 does not lead to epidermal barrier function (Kurbet *et al*, 2016). However, we are unable to further dissect the fate of the basal cells in cKO skin as the animals are neonatally lethal.

5. What are the main metabolic mechanism for dermal macrophages without induced inflammation? What metabolic change will happen if lactate is directly injected to the dermis in either embryos or adult mice?

The metabolic state of macrophages in the WT skin is an important and interesting question which we are actively pursuing this in our lab using a macrophage depleted mice model system. We have addressed the question of local lactate injection into the skin previously. We request the review to refer to our response to question No. 2.

6. Minor suggestions:

a) Metabolic pathway-related gene expression was tested and clustered. But for some key intermediate products, biochemical assays of quantification could be added for validation.

We have performed steady state metabolomics for the epidermis and dermis isolated from the cKO and WT skin to quantify the levels of glycolysis and TCA metabolites. The data for this is provided in Figure 1(D,E) and Figure 4 (C,D) in the main manuscript.

Epidermis Steady state metabolomics

Macrophage Steady state metabolomics

b) In several figures red shadows (or bushes) are used to depict ECM degradation, which can be confusing. Perhaps another way of illustration?

As suggested by the reviewer we have simplified the schematic representation of ECM degradation as shown below. We trust that will allay any confusion regarding the depiction of ECM degradation.

c) The effectiveness of treating pregnant dams with agents on the embryo could be discussed.

We are one of the few labs that have pioneering the methodology of administering drug in-utero and intraperitoneally to treat embryonic inflammation in pregnant dams as described in our previous papers (Bhattacharjee *et al*, 2021; Kurbet *et al*, 2016).

d) Provide more discussion on the implication of this finding for other autoimmune diseases.

As suggested by the reviewer, this is now been added in the discussion section of the revised manuscript.

Dear Dr Raghavan,

Thank you for submitting your revised manuscript (EMBOJ-2023-113774R-Q) to The EMBO Journal, as well as for your patience with our response at this time of the year. Your amended study was sent back to the two referees for their scientific re-evaluation, and we have received detailed comments from one of them, which I enclose below.

As you will see, the expert stated that the work has been substantially improved by the revisions and s/he is now in favour of publication. Please note that we have carefully tested your response to the other referee and found the concerns raised to be addressed satisfactorily.

Thus, we are pleased to inform you that your manuscript has been accepted in principle for publication in The EMBO Journal.

We now need you to take care of a number of issues related to formatting and data presentation as detailed below, which should be addressed at re-submission.

Please contact me at any time if you have additional questions related to below points.

As you might have seen on our web page, every paper at the EMBO Journal now includes a 'Synopsis', displayed on the html and freely accessible to all readers. The synopsis includes a 'model' figure as well as 2-5 one-short-sentence bullet points that summarize the article. I would appreciate if you could provide this figure and the bullet points.

Thank you for giving us the chance to consider your manuscript for The EMBO Journal. I look forward to your final revision.

Again, please contact me at any time if you need any help or have further questions.

Best regards,

Daniel Klimmeck

>> Please add up to five keywords for your study.

>> Adjust the title of the 'Competing Interests' section to 'Disclosure and Competing Interests Statement'.

>> Author Contributions: Please remove the author contributions information from the manuscript text. Note that CRediT has replaced the traditional author contributions section as of now because it offers a systematic machine-readable author contributions format that allows for more effective research assessment. and use the free text boxes beneath each contributing author's name to add specific details on the author's contribution.

More information is available in our guide to authors.

>> Figure callouts: are currently missing for Fig 4H and need to be provided.

>> Appendix file: All supplementary figures should be compiled in one PDF labelled "Appendix". The appendix should have a table of contents on the first page, with page numbers. The nomenclature should be corrected to "Appendix Figure S1" etc. Each appendix figure's legend should be added to the appendix file, underneath the corresponding figure. Figure S4a and 4b should be merged into one Appendix Figure S4.

>> Please include the funding information in the Acknowledgements section.

>> My colleague H. Sonntag (CC'ed in) will contact you shortly in a separate message to provide Source Data compiled as one

file per figure.

>> Data accessibility section: move to the end of Material and Methods section. Rename to 'Data availability section' and reference the RNAseq data published earlier. Please add: 'No additional data amenable to deposition in large-scale repositories were generated in this study.'

>> The "Once-Sentence-Summary" on p.2 can be removed.

>> The table with the drugs and treatments schedule should be moved after the references section; it should be named Table 1 and needs a short legend and a callout. The table with the antibody dilutions should be made Table 2, with the same adjustments as above, and the table with the primers should be Table 3.

>> Please provide a completed author checklist.

Referee #2:

The revised manuscript is significantly improved from the original version. The study is now more rigorous and the authors have satisfactorily answered the concerns initially raised.

The authors addressed the remaining minor editorial issues.

Dear Dr Srikala Raghavan,

Thank you for submitting the revised version of your manuscript. I have now evaluated your amended manuscript and concluded that the remaining minor concerns have been sufficiently addressed.

Thus, I am pleased to inform you that your manuscript has been accepted for publication in the EMBO Journal.

Please note that it is The EMBO Journal policy for the transcript of the editorial process (containing referee reports and your response letter) to be published as an online supplement to each paper. I would accordingly like to ask your consent on keeping the referee figure included in this file.

If you do NOT want the transparent process file published, you will need to inform the Editorial Office via email immediately. More information is available here: https://www.embopress.org/transparent-process#Review_Process

On a different note, I would like to alert you that EMBO Press offers a format for a video-synopsis of work published with us, which essentially is a short, author-generated film explaining the core findings in hand drawings, and, as we believe, can be very useful to increase visibility of the work. This has proven to offer a nice opportunity for exposure i.p. for the first author(s) of the study. Please see the following link for representative examples and their integration into the article web page:

<https://www.embopress.org/doi/full/10.15252/emj.2019103932>

Finally, we have noted that the submitted version of your article is also posted on the preprint platform bioRxiv. We would appreciate if you could alert bioRxiv on the acceptance of this manuscript at The EMBO Journal in order to allow for an update of the entry status. Thank you in advance!

If you have any questions, please do not hesitate to call or email the Editorial Office.

Best regards,

Daniel Klimmeck

Daniel Klimmeck, PhD
Senior Editor
The EMBO Journal
EMBO
Postfach 1022-40
Meyerohofstrasse 1
D-69117 Heidelberg

contact@embojournal.org
Submit at: <http://emboj.msubmit.net>
